# An Approach to Canonical Correlation Analysis Based on Rényi’s Pseudodistances

**DOI:** 10.3390/e25050713

**Published:** 2023-04-25

**Authors:** María Jaenada, Pedro Miranda, Leandro Pardo, Konstantinos Zografos

**Affiliations:** 1Interdisciplinary Mathematics Institute, Complutense University of Madrid, 28040 Madrid, Spain; 2Probability-Statistics and Operational Research Unit, Department of Mathematics, University of Ioannina, 45110 Ioannina, Greece

**Keywords:** Information Canonical Correlation Analysis, Kullback–Leibler divergence, mutual information, Renyi’s pseudodistances, robustness, consistency

## Abstract

Canonical Correlation Analysis (CCA) infers a pairwise linear relationship between two groups of random variables, X and Y. In this paper, we present a new procedure based on Rényi’s pseudodistances (RP) aiming to detect linear and non-linear relationships between the two groups. RP canonical analysis (RPCCA) finds canonical coefficient vectors, a and b, by maximizing an RP-based measure. This new family includes the Information Canonical Correlation Analysis (ICCA) as a particular case and extends the method for distances inherently robust against outliers. We provide estimating techniques for RPCCA and show the consistency of the proposed estimated canonical vectors. Further, a permutation test for determining the number of significant pairs of canonical variables is described. The robustness properties of the RPCCA are examined theoretically and empirically through a simulation study, concluding that the RPCCA presents a competitive alternative to ICCA with an added advantage in terms of robustness against outliers and data contamination.

## 1. Introduction

Canonical Correlation Analysis (CCA) is a statistical technique used to identify and measure associations among two sets of variables; in the following, denoted by $X_{q\times 1}$ and $Y_{p\times 1}$ (q≤p). It is appropriate in situations where multiple regression would be used but where there are multiple intercorrelated outcome variables. Hence, it allows us to summarize relationships into a lesser number of statistics while preserving the main facets of those relationships. CCA was first considered in [1] and has been widely used in the statistical literature; for example, to summarize relationships between sets of variables, to reduce the dimensionality of data or to transform two sets of variables into a new dataset of uncorrelated variables as a preprocessing step for the multiple linear regression model. More insight about CCA can be found, e.g., in [2,3].

CCA looks for two direction vectors a,b (canonical vectors) such that the linear combinations $U=a^{T}X$ and $V=b^{T}Y,$ so called canonical variables, are (linearly) correlated as much as possible. However, if a linear relationship does not exist between the pairs $a^{T}X$ and $b^{T}Y,$ CCA could fail in detecting these pairs of canonical vectors. In other words, CCA can only detect *linear* relations between the canonical variables, but other functional relationships may exist.

The linear restriction is a significant drawback of CCA when analyzing some real data with highly non-linear relationships. For example, Oulai et al. [4] presented a real situation with non-linear relationships between variables regarding the representation of a hydrological process in the delineation of homogeneous regions. In their context, the two groups of variables under consideration were hydrological variables and meteorological and/or graphical characteristics of watersheds, and their non-linear relationship depended essentially on the physiographic characteristics of the watersheds. Additionally, Ref. [5] presented a nice application of non-linear CCA to seasonal climate forecasting. In [6], some real life data with complex non-linear relationships that cannot be properly captivated by classical CCA are also presented. There is an extensive bibliography addressing non-linear CCA. Without wishing to cite all the existing literature on the topic, we would like to mention some interesting works on the subject: [7] (Chapter 6), [8,9,10,11] and references therein.

To shed light on this problem, let us consider the following situation described in [12]: let $X={\left(X_{1},X_{2}\right)}^{T}$ and $Y={\left(Y_{1},Y_{2}\right)}^{T}$ be a pair of random vectors such that
$$X∼N\left(0,I\right),\;Y_{1}=X_{1}^{2}+Z,\;Y_{2}=Z,$$
with $Z∼\chi_{1}^{2}$ and independent from X. In this case, $Cov\left(X,Y\right)=0_{2\times 2},$ and so the vectors X and Y are uncorrelated (so they are linearly independent). Consequently, classical CCA cannot detect that, indeed, $Y_{1}$ is related (although not linearly) to $X_{1},$ even if the variables are not *fully* independent. On the other hand, as the pair $\left(X,Y\right)$ does not follow a normal distribution (and therefore uncorrelation does not imply independence), a hidden relationship may exist (and indeed it does exist!) that has not been detected by CCA. Of course, under normality, rejecting any linear correlation using CCA implies independence between both variables.

It is not surprising that CCA fails in the previous example, as CCA focuses on “linear trends”, but the true relation underlying it is quadratic. To overcome this drawback, in a pioneer paper, Yin [12] proposed the use of the Kullback–Leibler divergence and developed a new procedure called *Informational Canonical Correlation Analysis* (ICCA), aiming to also detect non-linear relationships for linear combinations of the components.

Let $U=a^{T}X$ and $V=b^{T}Y$ be linear combinations of X and Y defining a pairwise of canonical variables, with $a\in R^{q}$ and $b\in R^{p}$. We denote by $f_{UV}\left(u,v\right)$ the joint probability density function (PDF) of (U,V), and, by $f_{U}\left(u\right)$ (resp. $f_{V}\left(v\right)$), the marginal unidimensional PDF of *U* (resp. *V*). From a statistical point of view, both canonical variables *U* and *V* would be independent if their joint distribution coincides with the product of the marginal PDF’s, $f_{UV}\left(u,v\right)=f_{U}\left(u\right)\times f_{V}\left(v\right),$ and, conversely, a strong dependence between *U* and *V* would result in a large statistical distance between the joint PDF and the product of the marginals. A suitable divergence should then be adopted to measure such statistical closeness of the two PDFs. The Kullback–Leibler divergence is the most commonly used measure for distinguishing two distributions, and it has a great statistical importance in the field of information theory.

The Kullback–Leibler divergence between $f_{UV}\left(u,v\right)$ and $f_{U}\left(u\right)\times f_{V}\left(v\right)$ is given as
(1)$$D_{KL}\left(a,b\right):=D_{KL}\left(f_{UV},f_{U}\times f_{V}\right)=\int_{R^{2}}f_{UV}\left(u,v\right)\ln\frac{f_{UV}\left(u,v\right)}{f_{U}\left(u\right)f_{V}\left(v\right)}dudv.$$The above divergence is not symmetric, so it quantifies the expected inaccuracy excess from using $f_{U}\times f_{V}$ as a model when the actual PDF is $f_{UV}.$ That is, the inaccuracy caused by assuming independence between the pair of canonical variables. Consequently, truly independent canonical variables should minimize the Kullback–Leibler divergence in Equation (Equation 1); conversely, functionally dependent canonical variables should maximize the divergence. For more details about the Kullback–Leibler divergence, see [13].

In this vein, ICCA aims to identify *q* pairwise canonical variables $a_{i}\in R^{q}$ and $b_{i}\in R^{p}$, i≤q≤p such that $D_{KL}\left(a_{i},b_{i}\right)=\max_{a,\;b}D_{KL}\left(a,b\right)$. However, the Kullback–Leibler divergence is invariant under linear transformations, and so there are infinitely many ways to define canonical vectors yielding the same objective function. Then, for identification, we constrain the canonical variables to have unit variance. Moreover, once a relationship is identified by a pair of canonical variables, we expect to exclude its effect from the consecutive canonical variables. For such a purpose, we also require that pairs of canonical variables are uncorrelatated with any other pair. That is, ICCA finds *q* linearly independent pairs of canonical variables with unit variance maximizing (in decreasing order) the corresponding Kullback–Leibler divergence. Mathematically, to compute each pair of variables, we need to solve the optimization problem $D_{KL}\left(a_{i},b_{i}\right)=\max_{a,\;b}D_{KL}\left(a,b\right)$ subject to $a_{i}^{T}\Sigma_{X}a_{i}=$$b_{i}^{T}\Sigma_{Y}b_{i}=1$ and $a_{j}^{T}\Sigma_{X}a_{i}=$$b_{j}^{T}\Sigma_{Y}b_{i}=0$ for j=1,…,i−1, where $\Sigma_{X}$ and $\Sigma_{Y}$ denote the variance–covariance matrices of X and Y, respectively. We apply the same for RP.

From Yin’s (2004) reinterpretation of the canonical analysis, several procedures based on divergence and entropy measures have been proposed to reduce the limitations of CCA. For example, Mandal et al. [14] considered $\left(\alpha ,\beta \right)$ divergence measures defined in [15], and Iaci and Sriram [6] used the density power divergence measures defined in [16] as a measure of statistical closeness. In [17], canonical dependence based on the squared-loss mutual information was studied. Other interesting results regarding ICCA can be seen in [18,19,20,21,22,23].

Despite its popularity, the Kullback–Leibler divergence association measure is quite sensitive to outlying observations, as pointed out in [24]. For outliers, we mean data that behave very differently to expectations according to the law modeling the relation. The main purpose of this paper is to extend the ICCA procedure to a wider family of robust methods based on RP divergence, which remains competitive to ICCA in terms of efficiency but provides a more stable estimation of the canonical vectors in the presence of contamination in the data.

The RP family, parameterized by a tuning parameter τ controlling the trade-off between robustness and efficiency, was considered for the first time in Jones et al. [25]. Later, Broniatowski et al. [26] demonstrated that RP is a proper divergence, positive for any two densities and for all values of the tuning parameter [26,27], and it is null if (and only if) both densities are the same. The theory in [26] for independent and identically distributed random variables was extended to the case of independent but not identically distributed random variables in [28]. They termed this family of pseudodistances as RP because of their similarities with Renyi’s divergence measures Rényi (1961) [29]. Rényi’s pseudodistance has shown promising behavior in other statistical problems, providing robust minimum RP estimators with good asymptotic and robustness properties, and it includes the Kullback–Leibler divergence as a particular case at τ=0. For example, Toma and Leoni-Aubin [30] considered efficient and robust measures for general parametric models based on RP and, Toma et al. [31] later developed a new criterion for model selection based on the RP. In [27], Castilla et al. introduced a family of Wald-type tests for testing the parameters in linear regression models, and these results were later extended for generalized linear regression models in [32,33]. Wald-type tests based on minimum RP estimators in bidimensional normal populations were considered in [34]. Jaenada et al. [35] introduced and studied the minimum RP estimators under restricted parameter spaces, which are of great statistical interest in many practical applications such as hypothesis testing. Under the name of *γ-entropy*, Fujisawa and Eguchi [36] applied RP to introduce robust estimators of general parametric families. Motivated for the great performance of the minimum RP estimator on those different statistical models in terms of robustness, we have adopted the RP divergence to extend the ICCA procedure.

The rest of the paper is organized as follows. The Rényi’s Pseudodistance Canonical Correlation Analysis (RPCCA) is introduced in Section 2, and some of its properties are studied. Next, an estimation design for computing the canonical vectors in practice using RPCCA is described in Section 3. In Section 4, the robustness of the RPCCA is theoretically established. Section 5 describes a permutation test to determine the number of significant canonical variables and thereby provide a dimension reduction method. In Section 6, a Monte Carlo simulation study is carried out to empirically evaluate the performance of the RPCCA and compare the proposed method with the ICCA in terms of estimation accuracy and robustness. An example with real data is studied in Section 6.3. Finally, some conclusions are drawn in Section 7.

## 2. Rényi’s Pseudodistance Canonical Correlation Analysis

Given two multidimensional random variables X and Y, the RPCCA aims to identify two direction vectors a and b (the canonical vectors), such that the corresponding canonical variables $U=a^{T}X$ and $V=b^{T}Y$ are as dependent as possible. Such dependency is measured in terms of RP between their joint distribution and the product of their marginal distributions. The **RP of tuning parameter τ** between the joint distribution of the bidimensional random variable (U,V) and the product of their marginals, $f_{U}\left(u\right)\times f_{V}\left(v\right)$, is given for τ>0 by (cf. [26]).
$$\begin{array}{ccc} d_{\tau }\left(a,b\right) & = & d_{\tau }\left(f_{U}\times f_{V},f_{UV}\right)\\ & = & \frac{1}{\tau +1}\ln\int_{R^{2}}f_{U}^{\tau +1}\left(u\right)f_{V}^{\tau +1}\left(v\right)dudv-\frac{1}{\tau }\ln\int_{R^{2}}f_{U}^{\tau }\left(u\right)f_{V}^{\tau }\left(v\right)f_{UV}\left(u,v\right)dudv\\ & & +\frac{1}{\tau \left(\tau +1\right)}\ln\int_{R^{2}}f_{UV}^{\tau +1}\left(u,v\right)dudv. \end{array}$$

Hence, the RP measures the statistical discrepancy between the joint PDF of the canonical variables, $f_{UV}$ and the marginal PDF’s product $f_{U}\times 4_{V},$ or, in other words, the loss in accuracy that comes with assuming independence.

For τ=0, the RP can be defined as the corresponding limit, τ→0, yielding the Kullback–Leibler divergence:(2)$$d_{0}\left(a,b\right)=\lim_{\tau ↓0}d_{\tau }\left(a,b\right)=\lim_{\tau ↓0}d_{\tau }\left(f_{U}\times f_{V},f_{UV}\right)=D_{KL}\left(f_{UV},f_{U}\times f_{V}\right).$$

As earlier discussed, independent canonical variables lead to $d_{\tau }\left(a,b\right)=0,$ and, contrarily, strong dependency should result in large RP distances. Then, the **RPCCA** procedure aims to identify pairwise canonical vectors $a_{i}\in R^{q}$ and $b_{i}\in R^{p}$, i≤q≤p such that
$$d_{\tau }\left(a_{i},b_{i}\right)=\max_{a,\;b}d_{\tau }\left(a,b\right),$$
and, as before for identification, the canonical variables should have unit variance and be uncorrelated with any previous pairwise of canonical variables:$$a_{i}^{T}\Sigma_{X}a_{i}=\;b_{i}^{T}\Sigma_{Y}b_{i}=1,\;\forall i,$$
$$a_{j}^{T}\Sigma_{X}a_{i}=\;b_{j}^{T}\Sigma_{Y}b_{i}=0,\;\forall j=1,\ldots ,i-1,$$
where $\Sigma_{X}$ and $\Sigma_{Y}$ are the variance–covariance matrices of X and Y, respectively.

Note that, by Equation (Equation 2), the ICCA procedure presented in [12] is recovered at τ=0, and so the RPCCA generalizes ICCA.

**Remark** **1.**
*Given the random vectors X and Y, RPCCA finds the vectors $a_{1},b_{1}$ such that $a_{1}^{T}X$ and $b_{1}^{T}Y$ are maximally related. This maximal relation is measured via $d_{\tau }\left(a_{i},b_{i}\right)$, as previously defined. Once these vectors $a_{1},b_{1}$ are obtained, the procedure looks for a new pair of vectors $a_{2},b_{2}$ such that $a_{1}$ and $a_{2}$ are incorrelated, and the same applies for $b_{1},b_{2}$, and $a_{2}^{T}X$ and $b_{2}^{T}Y$ are maximally related. Consequently,*

$$d_{\tau }\left(a_{1},b_{1}\right)\geq d_{\tau }\left(a_{2},b_{2}\right).$$

*Next, the procedure looks for $a_{3},b_{3}$ being incorrelated to $a_{1},a_{2}$ and $b_{1},b_{2},$ respectively, and so on. Hence, it follows that*

$$d_{\tau }\left(a_{i},b_{i}\right)\geq d_{\tau }\left(a_{i+1},b_{i+1}\right),$$

*for any i=1,…,q−1. If $d_{\tau }\left(a_{i},b_{i}\right)=0$. Then, independence arises and the procedure stops. In practice, we will have an estimation of $d_{\tau }\left(a_{i},b_{i}\right)$, and we will stop the procedure if this value does not exceed a certain threshold. This will be applied in Section 5 in order to determine the number of components.*


Let us consider, again, the example described in the introduction, where $X={\left(X_{1},X_{2}\right)}^{T}$ and $Y={\left(Y_{1},Y_{2}\right)}^{T}$ satisfy
$$X∼N\left(0,I\right),\;Y_{1}=X_{1}^{2}+Z,\;Y_{2}∼Z,\;Z∼\chi_{1}^{2}.$$The true value of the first pair of canonical vectors are then $a_{1}=b_{1}={\left(1,0\right)}^{T}.$ Under the described setup, it follows that
$$f_{UV}\left(u,v\right)=f_{X_{1},Y_{1}}\left(x,y\right)=\frac{1}{\pi }\frac{1}{2}\exp\left(-\frac{1}{2}y\right){\left(y-x^{2}\right)}^{-\frac{1}{2}},\;y>0,x\in R,$$
and
$$f_{U}\left(u\right)=f_{X_{1}}\left(x\right)=\frac{1}{\sqrt{2\pi }}\exp\left(-\frac{1}{2}x^{2}\right),\;f_{V}\left(v\right)=f_{Y_{1}}\left(y\right)=\frac{1}{2}\exp\left(-\frac{1}{2}y\right),\;y>0.$$Clearly,
$$f_{UV}\left(u,v\right)\neq f_{U}\left(u\right)\times f_{V}\left(v\right)$$
and because of the properties of the divergence
$$d_{\tau }\left(a_{1},b_{1}\right)>0.$$

The last inequality holds because the RP divergence, $d_{\tau }\left(\cdot ,\cdot \right),$ only reaches the value zero if both arguments coincide, as discussed in Section 1 (see [26] for more details). In this case, RPCCA should identify a pair $a_{1},b_{1}$ with a non-zero informational coefficient of canonical correlation defining the canonical variables $a_{1}^{T}X$ and $b_{1}^{T}Y$.

For practical use of RPCCA, it is interesting to note that RPCCA is equivariant under invertible linear transformations. This equality does not hold for other extensions of ICCA, but proportionality arises instead.

**Proposition** **1.**
*Consider two random variables U and V, and take R=cU and S=eV, where c and e are non-zero real numbers (Indeed, the result also holds if we consider two random vectors U,V, and consider R=CU and S=DV, where C and D are two invertible matrices. In this case, RP is computed considering multidimensional integrals.). Then,*

$$d_{\tau }\left(f_{U}\times f_{V},f_{UV}\right)=d_{\tau }\left(f_{R}\times f_{S},f_{RS}\right).$$



**Proof.** By definition,
$$\begin{array}{ccc} d_{\tau }\left(f_{R}\times f_{S},f_{RS}\right) & = & \frac{1}{\tau +1}\ln\int f_{R}^{\tau +1}\left(r\right)f_{S}^{\tau +1}\left(s\right)\;drds+\frac{1}{\tau (\tau +1)}\ln\int f_{RS}^{\tau +1}\left(r,s\right)\;drds\\ & & -\frac{1}{\tau }\ln\int f_{R}^{\tau }\left(r\right)f_{S}^{\tau }\left(s\right)f_{RS}\left(r,s\right)\;drd\;s\\ & = & \frac{1}{\tau +1}\ln\int f_{U}^{\tau +1}\left(u\right){\left(\frac{1}{c}\right)}^{\tau +1}f_{V}^{\tau +1}\left(v\right){\left(\frac{1}{e}\right)}^{\tau +1}ce\;dudv\\ & & +\frac{1}{\tau (\tau +1)}\ln\int f_{UV}^{\tau +1}\left(u,v\right){\left(\frac{1}{ce}\right)}^{\tau +1}ce\;dudv\\ & & -\frac{1}{\tau }\ln\int f_{U}^{\tau }\left(u\right){\left(\frac{1}{c}\right)}^{\tau }f_{V}^{\tau }\left(s\right){\left(\frac{1}{e}\right)}^{\tau }f_{UV}\left(u,v\right)\left(\frac{1}{ce}\right)ce\;dud\;v\\ & = & \frac{1}{\tau +1}\ln{\left(\frac{1}{\left|c||e\right|}\right)}^{\tau }\int f_{U}^{\tau +1}\left(u\right)f_{V}^{\tau +1}\left(v\right)\;dud\;v\\ & & +\frac{1}{\tau (\tau +1)}\ln{\left(\frac{1}{\left|c||e\right|}\right)}^{\tau }\int f_{UV}^{\tau +1}\left(u,v\right)\;dudv\\ & & -\frac{1}{\tau }\ln{\left(\frac{1}{\left|c||e\right|}\right)}^{\tau }\int f_{U}^{\tau }\left(u\right)f_{V}^{\tau }\left(v\right)f_{UV}\left(u,v\right)\;dud\;v\\ & = & \left(\frac{1}{\tau +1}+\frac{1}{\tau (\tau +1)}-\frac{1}{\tau }\right)\ln{\left(\frac{1}{\left|c||e\right|}\right)}^{\tau }+d_{\tau }\left(f_{U}\times f_{V},f_{UV}\right)\\ & = & d_{\tau }\left(f_{U}\times f_{V},f_{UV}\right). \end{array}$$□

The next result establishes that the RPCCA is reduced to CCA in the case of normal distributions.

**Proposition** **2.**
*In the case of normal distributions, RPCCA coincides with CCA.*


**Proof.** Consider normal populations, i.e., assume that the multidimensional random variables X and Y are jointly normally distributed,
$$\left(\begin{array}{c} X\\ Y \end{array}\right)\equiv N\left(\left(\begin{array}{c} \mu_{X}\\ \mu_{Y} \end{array}\right),\left(\begin{array}{cc} \Sigma_{X} & \Sigma_{XY}\\ \Sigma_{YX} & \Sigma_{Y} \end{array}\right)\right).$$Therefore, the bidimensional random variable $\left(U,V\right)=\left(a^{T}X,b^{T}Y\right)$ follows a bidimensional normal distribution whose vector mean is
$$\mu ={\left(\mu_{1},\mu_{2}\right)}^{T}={\left(E\left[a^{T}X\right],E\left[b^{T}Y\right]\right)}^{T}={\left(a^{T}\mu_{X},b^{T}\mu_{Y}\right)}^{T},$$
and the variance–covariance matrix is given by
$$\left(\begin{array}{cc} \sigma_{1}^{2} & \sigma_{1}\sigma_{2}\rho \\ \sigma_{1}\sigma_{2}\rho & \sigma_{2}^{2} \end{array}\right)$$
being
$$\sigma_{1}^{2}=Var\left[a^{T}X\right]=a^{T}\Sigma_{X}a,\;\sigma_{2}^{2}=Var\left[b^{T}Y\right]=b^{T}\Sigma_{Y}b\;\mathrm{and}\;\rho =\frac{Cov\left(U,V\right)}{\sigma_{1}\sigma_{2}}=\frac{a^{T}\Sigma_{XY}b}{\sigma_{1}\sigma_{2}}.$$On the other hand, the marginal densities $f_{\mu_{1},\sigma_{1}}\left(u\right)$ and $f_{\mu_{2},\sigma_{2}}\left(v\right)$ of $a^{T}X$ and $b^{T}Y,$ respectively, are normal distributions,
$$f_{U}\left(u\right)\equiv N\left(\mu_{1},\sigma_{1}^{2}\right)\;\mathrm{and}\;f_{V}\left(v\right)\equiv N\left(\mu_{2},\sigma_{2}^{2}\right).$$We first compute the RP between $f_{\mu_{1},\sigma_{1}}\left(u\right)\times$$f_{\mu_{2},\sigma_{2}}\left(v\right)$ and $f_{\mu_{1},\mu_{2},\sigma_{1,}\sigma_{2},\rho }\left(u,v\right).$ Considering the results obtained in Supplementary Materials (Appendix A) in [6], we have
$$\int_{R^{2}}f_{U}{\left(u\right)}^{\tau +1}f_{V}{\left(v\right)}^{\tau +1}dudv=k_{1}^{\tau }{\left(1+\tau \right)}^{-1},$$
being $k_{1}={\left(2\pi \sigma_{1}\sigma_{2}\right)}^{-1}$ and
$$\int_{R^{2}}f_{UV}{\left(u,v\right)}^{\tau +1}dudv=k_{1}^{\tau }{\left(1+\tau \right)}^{-1}{\left(1-\rho^{2}\right)}^{-\frac{\tau }{2}}.$$On the other hand, it is not difficult to see that
$$\int_{R^{2}}f_{U}^{\tau }\left(u\right)f_{V}^{\tau }\left(v\right)f_{UV}\left(u,v\right)dudv=k_{1}^{\tau }{\left[(1+\tau (1+\rho ))(1+\tau (1-\rho ))\right]}^{-1/2}.$$Based on the previous quantities, we have
$$\begin{array}{ccc} d_{\tau }\left(a,b\right) & = & \frac{1}{\tau +1}\ln k_{1}^{\tau }{\left(1+\tau \right)}^{-1}-\frac{1}{\tau }\ln k_{1}^{\tau }k_{1}^{\tau }{\left[(1+\tau (1+\rho ))(1+\tau (1-\rho ))\right]}^{-1/2}\\ & & +\frac{1}{\tau (\tau +1)}\ln k_{1}^{\tau }{\left(1+\tau \right)}^{-1}{\left(1-\rho^{2}\right)}^{-\tau /2}\\ & = & \ln\frac{{\left((1+\tau (1+\rho ))(1+\tau (1-\rho ))\right)}^{1/2\tau }}{{\left(1+\tau \right)}^{1/\tau }{\left(1-\rho^{2}\right)}^{1/2(\tau +1)}}. \end{array}$$For fixed τ, it can be seen from the previous expression that $d_{\tau }\left(a,b\right)$ depends on ρ. Moreover, it is not difficult to show that $d_{\tau }\left(a,b\right)$ is an increasing function on $\rho^{2}$ for any τ (see Figure 1 for τ=0.1,0.3 and 0.9). To show this, it suffices to see that
$$f_{\tau }\left(\rho \right)=\frac{{\left[(1+\tau (1+\rho ))(1+\tau (1-\rho ))\right]}^{1/2\tau }}{{\left(1+\tau \right)}^{1/\tau }{\left(1-\rho^{2}\right)}^{1/2(\tau +1)}},\rho \in \left(-1,1\right)$$
is increasing in $\rho^{2}$, and so it will be its logarithm transform. Now, note that
$$\left(1+\tau \left(1+\rho \right)\right)\left(1+\tau \left(1-\rho \right)\right)=1+2\tau +\tau^{2}\left(1-\rho^{2}\right).$$So, it suffices to show that the function
$$\frac{{\left(1+2\tau +\tau^{2}\left(1-\rho^{2}\right)\right)}^{1/2\tau }}{{\left(1-\rho^{2}\right)}^{1/2(\tau +1)}}$$
is increasing in $\rho^{2}.$ Taking derivatives with respect to $\rho^{2}$, we obtain
$$f^{\prime }\left(\rho^{2}\right)=\left[\frac{1}{2\tau }{\left(1+2\tau +\tau^{2}\left(1-\rho^{2}\right)\right)}^{1/2\tau -1}\left(-\tau^{2}\right){\left(1-\rho^{2}\right)}^{1/2(\tau +1)}\right.$$
$$-\left.\frac{1}{2(\tau +1)}{\left(1-\rho^{2}\right)}^{1/2(\tau +1)-1}{\left(1+2\tau +\tau^{2}\left(1-\rho^{2}\right)\right)}^{1/2\tau }\right]\frac{1}{{\left(1-\rho^{2}\right)}^{\frac{1}{\tau +1}}}.$$Thus, it suffices to check the non-negativity of
$$\left[\frac{1}{2\tau }\left(-\tau^{2}\right)\left(1-\rho^{2}\right)+\frac{1}{2(\tau +1)}\left(1+2\tau +\tau^{2}\left(1-\rho^{2}\right)\right)\right].$$Finally,
$$\begin{array}{cc} \; & \left[\frac{1}{2\tau }\left(-\tau^{2}\right)\left(1-\rho^{2}\right)+\frac{1}{2(\tau +1)}\left(1+2\tau +\tau^{2}\left(1-\rho^{2}\right)\right)\right]=\\ \; & \frac{-\tau^{2}\left(1-\rho^{2}\right)}{2\tau (\tau +1)}+\frac{1+2\tau }{2(\tau +1)}=\frac{\tau^{2}\left(\rho^{2}+1\right)+\tau }{2\tau (\tau +1)}>0. \end{array}$$
and the result holds. Thus, RPCCA is equivalent to classical CCA in the case of random normal variables. □

It can be seen that $d_{\tau }\left(a;b\right)$ is an increasing function on $\rho^{2}$ for any τ>0 under normal distributions; hence, RPCCA also extends CCA with a tuning parameter τ determining the sharpness of the distance $d_{\tau }\left(a,b\right)$ (or the function $f_{\tau }\left(\cdot \right)$ in the proof of Proposition 2).

## 3. Consistency

We now focus on the practical side of the RPCCA estimation. In practice, the PDFs $f_{UV},$$f_{U}$ and $f_{V}$ are unknown; thus, they should be empirically estimated. Likewise, the RPCCA should be formulated for an empirical setup.

The RP, $d_{\tau }\left(a,b\right),$ can be expressed in terms of expected values as
(3)$$\begin{array}{cc} d_{\tau }\left(a,b\right)= & \frac{1}{\tau \left(\tau +1\right)}\ln E_{f_{UV}}\left[f_{U,V}{\left(U,V\right)}^{\tau }\right]-\frac{1}{\tau }\ln E_{f_{UV}}\left[f_{U}^{\tau }\left(U\right)f_{V}^{\tau }\left(V\right)\right]\\ \; & +\frac{1}{\tau +1}\ln E_{f_{U}}\left[f_{U}{\left(U\right)}^{\tau }\right]E_{f_{V}}\left[f_{V}\left(V\right)\right]. \end{array}$$This interpretation of $d_{\tau }\left(a,b\right)$ makes the definition of its empirical estimator easier. Let $\left(X_{i},Y_{i}\right),i=1,\ldots ,n$ be a random sample of size *n* from the multidimensional random variables $\left(X,Y\right).$ Then, an empirical estimator of $d_{\tau }\left(a,b\right)$ is given by
(4)$$\begin{array}{ccc} \hat{d_{\tau }^{n}}\left(a,b\right) & = & \frac{1}{\tau \left(\tau +1\right)}\ln\left[\frac{1}{n}\sum_{i=1}^{n}{\hat{f_{UV}^{n}}}^{\tau }\left(u_{i},v_{i}\right)\right]-\frac{1}{\tau }\ln\left[\frac{1}{n}\sum_{i=1}^{n}{\hat{f_{U}^{n}}}^{\tau }\left(u_{i}\right){\hat{f_{V}^{n}}}^{\tau }\left(v_{i}\right)\right] \end{array}$$
(5)$$\begin{array}{ccc} & & +\frac{1}{\tau +1}\ln\left[\frac{1}{n}\sum_{i=1}^{n}{\hat{f_{U}^{n}}}^{\tau }\left(u_{i}\right)\frac{1}{n}\sum_{i=1}^{n}{\hat{f_{V}^{n}}}^{\tau }\left(v_{i}\right)\right]. \end{array}$$

Here, $\hat{f_{U}^{n}}\left(u\right),$$\hat{f_{V}^{n}}\left(v\right)$ and $\hat{f_{UV}^{n}}\left(u,v\right)$ are kernel density estimators of $f_{U}\left(u\right),$$f_{V}\left(v\right)$ and $f_{UV}\left(u,v\right),$ respectively, given by
(6)$$\hat{f_{U}^{n}}\left(u\right)=\frac{1}{na_{n}^{1}}\sum_{i=1}^{n}K\left(\frac{u-u_{i}}{a_{n}^{1}}\right),\;u\in R,$$
(7)$$\hat{f_{V}^{n}}\left(v\right)=\frac{1}{na_{n}^{2}}\sum_{i=1}^{n}K\left(\frac{v-v_{i}}{a_{n}^{2}}\right),\;v\in R,$$
and
(8)$$\hat{f_{UV}^{n}}\left(u,v\right)=\frac{1}{nb_{n}^{1}b_{n}^{2}}\sum_{i=1}^{n}K\left(\frac{u-u_{i}}{b_{n}^{1}}\right)K\left(\frac{v-v_{i}}{b_{n}^{2}}\right).$$

For the PDF’s estimators, we will use the univariate Gaussian kernel with $a_{n}^{j}=1.06n^{-0.2}s_{j}$ and $b_{n}^{j}=n^{-1/6}s_{j}$ for j=1,2, and the corresponding sample standard deviations $s_{1}$ and $s_{2}.$ This kernel function was proposed in [37] and adopted in many other extensions of ICCA, but other types of kernels could be considered instead, as long as they satisfy the conditions of Lemma 1 below (When the distribution is known up to a parameter value $f_{\theta },$ this information should be taken into account. Hence, the procedure would be the usual procedure in these situations. First, we estimate the parameter of the distribution θ by $\hat{\theta }$ and then consider the distribution with the estimated parameters $f_{\hat{\theta }}$. Next, we use $f_{\hat{\theta }}$ instead of $\hat{f}.$). Other interesting results about kernel distributions can be found in [38,39].

Then, the estimated canonical vectors, based on the RP with tuning parameter τ can be computed as
(9)$$\begin{array}{cc} \; & \left({\hat{a}}_{n}^{\tau },{\hat{b}}_{n}^{\tau }\right)=\arg\max_{a,b}\hat{d_{\tau }^{n}}\left(a,b\right),\\ \; & \;s.t.\;{\left(a_{n}^{\tau }\right)}^{T}{\hat{\Sigma }}_{11}a_{n}^{\tau }=1\;\mathrm{and}\;{\left(b_{n}^{\tau }\right)}^{T}{\hat{\Sigma }}_{22}b_{n}^{\tau }=1, \end{array}$$
where ${\hat{\Sigma }}_{11}$ and ${\hat{\Sigma }}_{22}$ are the empirical estimators of the variance–covariance matrices of X and Y, respectively.

We next establish the consistency of the estimated canonical vectors under some regularity conditions. That is, we will prove that the estimated canonical vectors $\left({\hat{a}}_{n}^{\tau },{\hat{b}}_{n}^{\tau }\right)$ converge for large sample sizes to the true canonical vectors defining the underlying functional relationship. For such a result, it is necessary to present the following lemma whose proof can be found in [40].

**Lemma** **1.**
*Let $(X_{i},Y_{i}),\;i=1,\ldots ,n$ be i.i.d. replications of the multidimensional random variables (X,Y). Consider a sequence ${\left\{a_{n}\right\}}_{n\in N}$ such that $0<a_{n}$ and $\lim_{n\to \infty }a_{n}=0.$ Assume*

$$\sum_{n=1}^{\infty }e^{-\gamma na_{n}^{2}}<\infty ,\sum_{n=1}^{\infty }e^{-\gamma na_{n}^{4}}<\infty ,\forall \gamma >0.$$


*Consider a function K of bounded variation (Consider a function $g:R^{k}\mapsto R$, and let P be the set of finite partitions of $R^{k}$ in rectangles $p=\{\left[x_{j},y_{j}\right),j=1,\ldots ,u_{p}\}$. Then, g is said to be of bounded variation if $sup_{p\in P}\{\sum_{j=1}^{u_{p}}\sum_{\epsilon_{1},\ldots ,\epsilon_{k}\in {\left\{0,1\right\}}^{k}}{\left(-1\right)}^{\sum_{i=1}^{k}\epsilon_{i}}g(\epsilon_{1}x_{j1}+\left(1-\epsilon_{1}\right)y_{j1},\ldots ,\epsilon_{k}x_{jk}+$$\left(1-\epsilon_{k}\right)y_{jk})\}<\infty .$) and suppose $f_{U}\left(a^{T}x\right)$ is uniformly continuous in a and $x,f_{V}\left(b^{T}y\right)$ is uniformly continuous in b and y, and $f_{UV}\left(a^{T}x,b^{T}y\right)$ is uniformly continuous in a,x,b and y. Then,*

$$\underset{a,x}{\sup}\left|\hat{f_{U}^{n}}\left(a^{T}x\right)-f_{U}\left(a^{T}x\right)\right|\overset{a.s.}{⟶}0.$$


$$\underset{b,y}{\sup}\left|\hat{f_{V}^{n}}\left(b^{T}y\right)-f_{V}\left(b^{T}y\right)\right|\overset{a.s.}{⟶}0.$$


$$\underset{a,b,x,y}{\sup}\left|\hat{f_{UV}^{n}}\left(a^{T}x,b^{T}y\right)-f_{UV}\left(a^{T}x,b^{T}y\right)\right|\overset{a.s.}{⟶}0.$$



Note that the Gaussian kernel functions defined in Equations (Equation 6)–(Equation 8) satisfy the conditions of Lemma 1. Of course, any other election of the kernel should also satisfy these regularity conditions. Now, let us define for any real value b>0 the set of indices such that the observations $a^{T}x_{i}$ and $b^{T}y_{i},i=1,\ldots ,n$ have positive densities
$$\chi_{b}=\left\{i:f_{UV}^{\tau }\left(a^{T}x_{i},b^{T}y_{i}\right)\geq b,\;f_{U}^{\tau }\left(a^{T}x_{i}\right)\geq b,\;f_{V}^{\tau }\left(b^{T}y_{i}\right)\geq b\right\}$$
and denote by $n_{b}$ the number of data outside this set. The next result establishes the consistency of the RPCCA.

**Proposition** **3.**
*Suppose the conditions of Lemma 1 hold. Assume b→0 such that*

$$\frac{n_{b}}{n}\underset{n\to \infty }{\overset{P}{⟶}}0,$$

*and consider the estimated and true pairs of canonical vectors,*

$$\left({\hat{a}}_{n},{\hat{b}}_{n}\right)=\arg\max_{a,b}\hat{d_{\tau }^{n}}\left(a,b\right)\;\mathrm{and}\;\left(a^{*},b^{*}\right)=\arg\max_{a,b}d_{\tau }\left(a,b\right).$$

*Further, assume that the maximum $\left(a^{*},b^{*}\right)$ is unique. Then,*

$$\left({\hat{a}}_{n},{\hat{b}}_{n}\right)\underset{n\to \infty }{\overset{P}{⟶}}\left(a^{*},b^{*}\right).$$



**Proof.** Take 0<ϵ,0<b such that $\epsilon \underset{n\to \infty }{⟶}0,b\underset{n\to \infty }{⟶}0$ and $\epsilon b^{-1}\underset{n\to \infty }{⟶}0.$By identification, we can assume ${\hat{a}}_{n}^{T}\Sigma_{11}{\hat{a}}_{n}={\hat{b}}_{n}^{T}\Sigma_{22}{\hat{b}}_{n}=1.$ Let us suppose that
$$\left({\hat{a}}_{n},{\hat{b}}_{n}\right)\underset{n\to \infty }{\overset{P}{↛}}\left(a^{*},b^{*}\right).$$Hence, there exists a subsequence of $\left\{\left({\hat{a}}_{n},{\hat{b}}_{n}\right)\right\}$ (that will be denoted by $\left({\hat{a}}_{n},{\hat{b}}_{n}\right)$ to avoid hard notation) and $\left(a_{0},b_{0}\right)$ such that $a_{0}^{T}\Sigma_{11}a_{0}=b_{0}^{T}\Sigma_{22}b_{0}=1,\left(a_{0},b_{0}\right)\neq \left(a^{*},b^{*}\right)$ and
$$\left({\hat{a}}_{n},{\hat{b}}_{n}\right)⟶\left(a_{0},b_{0}\right).$$Now, applying Lemma 1, we know that
$$\sup|\hat{f_{U}^{n}}\left({\hat{a}}_{n}^{T}x_{i}\right)-f_{U}\left({\hat{a}}_{n}^{T}x_{i}\right)|\underset{n\to \infty }{\overset{a.s.}{⟶}}0.$$
$$\sup|\hat{f_{V}^{n}}\left({\hat{b}}_{n}^{T}y_{i}\right)-f_{V}\left({\hat{b}}_{n}^{T}y_{i}\right)|\underset{n\to \infty }{\overset{a.s.}{⟶}}0.$$
$$\sup|\hat{f_{UV}^{n}}\left({\hat{a}}_{n}^{T}x_{i},{\hat{b}}_{n}^{T}y_{i}\right)-f_{UV}\left({\hat{a}}_{n}^{T}x_{i},{\hat{b}}_{n}^{T}y_{i}\right)|\underset{n\to \infty }{\overset{a.s.}{⟶}}0.$$Thus, for τ>0,
$$\sup|{\hat{f_{U}^{n}}}^{\tau }\left({\hat{a}}_{n}^{T}x_{i}\right)-f_{U}^{\tau }\left({\hat{a}}_{n}^{T}x_{i}\right)|\underset{n\to \infty }{\overset{a.s.}{⟶}}0.$$
$$\sup|{\hat{f_{V}^{n}}}^{\tau }\left({\hat{b}}_{n}^{T}y_{i}\right)-f_{V}^{\tau }\left({\hat{b}}_{n}^{T}y_{i}\right)|\underset{n\to \infty }{\overset{a.s.}{⟶}}0.$$
$$\sup|{\hat{f_{UV}^{n}}}^{\tau }\left({\hat{a}}_{n}^{T}x_{i},{\hat{b}}_{n}^{T}y_{i}\right)-f_{UV}^{\tau }\left({\hat{a}}_{n}^{T}x_{i},{\hat{b}}_{n}^{T}y_{i}\right)|\underset{n\to \infty }{\overset{a.s.}{⟶}}0.$$Hence, for an *n* large enough,
$${\hat{f_{U}^{n}}}^{\tau }\left({\hat{a}}_{n}^{T}x_{i}\right)=f_{U}^{\tau }\left({\hat{a}}_{n}^{T}x_{i}\right)+\Delta_{1i}=f_{U}^{\tau }\left(a_{0}^{T}x_{i}\right)+\delta_{1i},$$
$${\hat{f_{V}^{n}}}^{\tau }\left({\hat{b}}_{n}^{T}y_{i}\right)=f_{V}^{\tau }\left({\hat{b}}_{n}^{T}y_{i}\right)+\Delta_{2i}=f_{V}^{\tau }\left(b_{0}^{T}y_{i}\right)+\delta_{2i},$$
(10)$${\hat{f_{UV}^{n}}}^{\tau }\left({\hat{a}}_{n}^{T}x_{i},{\hat{b}}_{n}^{T}y_{i}\right)=f_{UV}^{\tau }\left({\hat{a}}_{n}^{T}x_{i},{\hat{b}}_{n}^{T}y_{i}\right)+\Delta_{3i}=f_{UV}^{\tau }\left(a_{0}^{T}x_{i},b_{0}^{T}y_{i}\right)+\delta_{3i}.$$Here, $|\delta_{1i}\left|,\right|\delta_{2i}\left|,\right|\delta_{3i}|<\epsilon .$ Remark that $\ln\left(\frac{1}{n}\sum_{i=1}^{n}{\hat{f_{UV}^{n}}}^{\tau }\left({\hat{a}}_{n}^{T}x_{i},{\hat{b}}_{n}^{T}y_{i}\right)\right)$ can be written as
$$\ln\left(\frac{1}{n}\sum_{i=1}^{n}{\hat{f_{UV}^{n}}}^{\tau }\left({\hat{a}}_{n}^{T}x_{i},{\hat{b}}_{n}^{T}y_{i}\right)\frac{\frac{1}{n}\sum_{i=1}^{n}f_{UV}^{\tau }\left(a_{0}^{T}x_{i},b_{0}^{T}y_{i}\right)}{\frac{1}{n}\sum_{i=1}^{n}f_{UV}^{\tau }\left(a_{0}^{T}x_{i},b_{0}^{T}y_{i}\right)}\right)$$
$$=\ln\left(\frac{1}{n}\sum_{i=1}^{n}f_{UV}^{\tau }\left(a_{0}^{T}x_{i},b_{0}^{T}y_{i}\right)\right)+\ln\left(\frac{\frac{1}{n}\sum_{i=1}^{n}{\hat{f_{UV}^{n}}}^{\tau }\left({\hat{a}}_{n}^{T}x_{i},{\hat{b}}_{n}^{T}y_{i}\right)}{\frac{1}{n}\sum_{i=1}^{n}f_{UV}^{\tau }\left(a_{0}^{T}x_{i},b_{0}^{T}y_{i}\right)}\right).$$Now, applying Equation (Equation 10), we obtain
$$\frac{\frac{1}{n}\sum_{i=1}^{n}{\hat{f_{UV}^{n}}}^{\tau }\left({\hat{a}}_{n}^{T}x_{i},{\hat{b}}_{n}^{T}y_{i}\right)}{\frac{1}{n}\sum_{i=1}^{n}f_{UV}^{\tau }\left(a_{0}^{T}x_{i},b_{0}^{T}y_{i}\right)}=1+\frac{\frac{1}{n}\sum_{i=1}^{n}\delta_{3i}}{\frac{1}{n}\sum_{i=1}^{n}f_{UV}^{\tau }\left(a_{0}^{T}x_{i},b_{0}^{T}y_{i}\right)}.$$The same can be performed for the two other cases. As $|\delta_{3i}|<\epsilon ,$ it follows that
$$\frac{1}{n}\sum_{i=1}^{n}\delta_{3i}\leq \epsilon .$$On the other hand,
$$\frac{1}{n}\sum_{i=1}^{n}f_{UV}^{\tau }\left(a_{0}^{T}x_{i},b_{0}^{T}y_{i}\right)\geq \frac{1}{n}\sum_{i=1}^{n}I\left(i\in \chi_{b}\right)f_{UV}^{\tau }\left(a_{0}^{T}x_{i},b_{0}^{T}y_{i}\right)\geq \frac{n-n_{b}}{n}b.$$As $\epsilon b^{-1}\to 0$ and $\frac{n_{b}}{n}\to 0,$ we conclude that
$$\frac{\frac{1}{n}\sum_{i=1}^{n}\delta_{3i}}{\frac{1}{n}\sum_{i=1}^{n}f_{UV}^{\tau }\left(a_{0}^{T}x_{i},b_{0}^{T}y_{i}\right)}\to 0.$$The same can be performed for the two other cases. Hence,
$$\begin{array}{ccc} {\hat{d}}_{\tau }^{n}\left({\hat{a}}_{n},{\hat{b}}_{n}\right) & = & \frac{1}{\tau +1}\ln\left(\frac{1}{n^{2}}\sum_{i=1}^{n}f_{U}^{\tau }\left(a_{0}^{T}x_{i}\right)\sum_{i=1}^{n}f_{V}^{\tau }\left(b_{0}^{T}y_{i}\right)\right)+\frac{1}{\tau +1}o\left(1\right)\\ & & +\frac{1}{\tau (\tau +1)}\ln\left(\frac{1}{n}\sum_{i=1}^{n}f_{UV}^{\tau }\left(a_{0}^{T}x_{i},b_{0}^{T}y_{i}\right)\right)+\frac{1}{\tau (\tau +1)}o\left(1\right)\\ & & -\frac{1}{\tau }\ln\left(\frac{1}{n}\sum_{i=1}^{n}f_{U}^{\tau }\left(a_{0}^{T}x_{i}\right)f_{V}^{\tau }\left(b_{0}^{T}y_{i}\right)\right)-\frac{1}{\tau }o\left(1\right)\\ & = & {\bar{d}}_{\tau }^{n}\left(a_{0},b_{0}\right)+o\left(1\right), \end{array}$$
with
$$\begin{array}{ccc} {\bar{d}}_{\tau }^{n}\left(a_{0},b_{0}\right) & = & \frac{1}{\tau +1}\ln\left(\frac{1}{n^{2}}\sum_{i=1}^{n}f_{U}^{\tau }\left(a_{0}^{T}x_{i}\right)\sum_{i=1}^{n}f_{V}^{\tau }\left(b_{0}^{T}y_{i}\right)\right)\\ & & +\frac{1}{\tau (\tau +1)}\ln\left(\frac{1}{n}\sum_{i=1}^{n}f_{UV}^{\tau }\left(a_{0}^{T}x_{i},b_{0}^{T}y_{i}\right)\right)\\ & & -\frac{1}{\tau }\ln\left(\frac{1}{n}\sum_{i=1}^{n}f_{U}^{\tau }\left(a_{0}^{T}x_{i}\right)f_{V}^{\tau }\left(b_{0}^{T}y_{i}\right)\right). \end{array}$$Note that ${\bar{d}}_{\tau }^{n}\left(a_{0},b_{0}\right)-d_{\tau }\left(a_{0},b_{0}\right)$ is given by
$$\frac{1}{\tau (\tau +1)}\ln\left(\frac{1}{n}\sum_{i=1}^{n}f_{UV}^{\tau }\left(a_{0}^{T}x_{i},b_{0}^{T}y_{i}\right)\right)-\frac{1}{\tau (\tau +1)}\ln\left(E_{f_{UV}}\left(f_{UV}^{\tau }\left(a_{0}^{T}X,b_{0}^{T}Y\right)\right)\right)$$
$$+\frac{1}{\left(\tau +1\right)}\ln\left(\frac{1}{n^{2}}\sum_{i=1}^{n}f_{U}^{\tau }\left(a_{0}^{T}x_{i}\right)\sum_{i=1}^{n}I\left(i\in \chi_{b}\right)f_{V}^{\tau }\left(b_{0}^{T}y_{i}\right)\right)-\frac{1}{\left(\tau +1\right)}\ln\left(E_{f_{U}}f_{U}^{\tau }\left(a_{0}^{T}X\right)E_{f_{V}}\left(f_{V}^{\alpha }\left(b_{0}^{T}Y\right)\right)\right)$$
$$-\frac{1}{\tau }\ln\left(\frac{1}{n}\sum_{i=1}^{n}f_{U}^{\tau }\left(a_{0}^{T}x_{i}\right)f_{V}^{\tau }\left(b_{0}^{T}y_{i}\right)\right)+\frac{1}{\tau }\ln\left(E_{f_{UV}}\left(f_{U}^{\tau }\left(a_{0}^{T}X\right)f_{V}^{\tau }\left(b_{0}^{T}Y\right)\right)\right).$$As ln is continuous and applying the Strong Law of Large Numbers, it follows
$$\ln\left(\frac{1}{n}\sum_{i=1}^{n}f_{UV}^{\tau }\left(a_{0}^{T}x_{i},b_{0}^{T}y_{i}\right)\right)\underset{n\to \infty }{\overset{a.s.}{⟶}}\ln\left(E_{f_{UV}}\left(f_{UV}^{\tau }\left(a_{0}^{T}X,b_{0}^{T}Y\right)\right)\right).$$We can perform this similarly for the two other lines. We conclude that ${\bar{d}}_{\tau }^{n}\left(a_{0},b_{0}\right)\underset{n\to \infty }{\overset{P}{⟶}}d_{\tau }\left(a_{0},b_{0}\right)$, and hence ${\hat{d}}_{\tau }^{n}\left(a_{0},b_{0}\right)\underset{n\to \infty }{\overset{P}{⟶}}d_{\tau }\left(a_{0},b_{0}\right).$ On the other hand, $\hat{d_{\tau }^{n}}\left({\hat{a}}_{n},{\hat{b}}_{n}\right)\geq \hat{d_{\tau }^{n}}\left(a^{*},b^{*}\right)$ because $\left({\hat{a}}_{n},{\hat{b}}_{n}\right)$ is the optimum by definition.Taking limits,
$$d_{\tau }\left(a_{0},b_{0}\right)=\lim_{n\to \infty }\hat{d_{\tau }^{n}}\left({\hat{a}}_{n},{\hat{b}}_{n}\right)\geq \lim_{n\to \infty }\hat{d_{\tau }^{n}}\left(a^{*},b^{*}\right)=d_{\tau }\left(a^{*},b^{*}\right).$$However, $d_{\tau }\left(a^{*},b^{*}\right)\geq d_{\tau }\left(a_{0},b_{0}\right)$ because $\left(a^{*},b^{*}\right)$ is the optimum. Hence, as $\left(a^{*},b^{*}\right)$ is the only maximum, we conclude that
$$\left(a^{*},b^{*}\right)=\left(a_{0},b_{0}\right),$$
a contradiction. □

## 4. Robustness

To motivate the inherent robustness property of the RPCCA procedure, we examine the behavior of the estimated divergence in Equation (Equation 4) for small values of the tuning parameter. The presented heuristic argument was first discussed in [6] for the density power divergence generalization of ICCA. Consider the estimated RP
$$\begin{array}{ccc} \hat{d_{\tau }^{n}}\left(a,b\right) & := & \frac{1}{\tau +1}\ln\left[\left(\frac{1}{n}\sum_{i=1}^{n}{\hat{f_{U}^{n}}}^{\tau }\left(u_{i}\right)\right)\left(\frac{1}{n}\sum_{i=1}^{n}{\hat{f_{V}^{n}}}^{\tau }\left(v_{i}\right)\right)\right]-\frac{1}{\tau }\ln\left[\frac{1}{n}\sum_{i=1}^{n}{\hat{f_{U}^{n}}}^{\tau }\left(u_{i}\right){\hat{f_{V}^{n}}}^{\tau }\left(v_{i}\right)\right]\\ & & +\frac{1}{\tau (\tau +1)}\ln\left[\frac{1}{n}\sum_{i=1}^{n}{\hat{f_{UV}^{n}}}^{\tau }\left(u_{i},v_{i}\right)\right]. \end{array}$$
and let the tuning parameter be τ↓0. Taking limits in the estimated divergence defined in Equation (Equation 4), the first term vanishes, and therefore
$$\hat{d_{\tau }^{n}}\left(a,b\right)\approx -\frac{1}{\tau }\ln\left[\frac{1}{n}\sum_{i=1}^{n}{\hat{f_{U}^{n}}}^{\tau }\left(u_{i}\right){\hat{f_{V}^{n}}}^{\tau }\left(v_{i}\right)\right]+\frac{1}{\tau (\tau +1)}\ln\left[\frac{1}{n}\sum_{i=1}^{n}{\hat{f_{UV}^{n}}}^{\tau }\left(u_{i},v_{i}\right)\right].$$We first study the limiting behavior of the first term,
$$l_{\tau }:=-\frac{1}{\tau }\ln\left[\frac{1}{n}\sum_{i=1}^{n}{\hat{f_{U}^{n}}}^{\tau }\left(u_{i}\right){\hat{f_{V}^{n}}}^{\tau }\left(v_{i}\right)\right].$$For τ↓0, this term is a limit of indeterminate form (0/0). Applying L’Hôpital’s rule, we obtain
$$l_{\tau }\approx \frac{\frac{1}{n}\left[\sum_{i=1}^{n}{\hat{f_{U}^{n}}}^{\tau }\left(u_{i}\right){\hat{f_{V}^{n}}}^{\tau }\left(v_{i}\right)\ln\left(\hat{f_{U}^{n}}\left(u_{i}\right)\right)+{\hat{f_{V}^{n}}}^{\tau }\left(v_{i}\right){\hat{f_{U}^{n}}}^{\tau }\left(u_{i}\right)\ln\left(\hat{f_{V}^{n}}\left(v_{i}\right)\right)\right]}{\frac{1}{n}\sum_{i=1}^{n}{\hat{f_{U}^{n}}}^{\tau }\left(u_{i}\right){\hat{f_{V}^{n}}}^{\tau }\left(v_{i}\right)}.$$Now, the denominator tends to 1 when τ↓0 so that
$$l_{\tau }\approx \frac{1}{n}\left[\sum_{i=1}^{n}{\hat{f_{U}^{n}}}^{\tau }\left(u_{i}\right){\hat{f_{V}^{n}}}^{\tau }\left(v_{i}\right)\ln\left(\hat{f_{U}^{n}}\left(u_{i}\right)\right)+{\hat{f_{V}^{n}}}^{\tau }\left(v_{i}\right){\hat{f_{U}^{n}}}^{\tau }\left(u_{i}\right)\ln\left(\hat{f_{V}^{n}}\left(v_{i}\right)\right)\right].$$

Similarly, consider
$$m_{\tau }:=\frac{1}{\tau (\tau +1)}\ln\left[\frac{1}{n}\sum_{i=1}^{n}{\hat{f_{UV}^{n}}}^{\tau }\left(u_{i},v_{i}\right)\right]$$
and consider its L’Hôpital approximation given by
$$m_{\tau }\approx \frac{1}{2\tau +1}\frac{\frac{1}{n}\sum_{i=1}^{n}{\hat{f_{UV}^{n}}}^{\tau }\left(u_{i},v_{i}\right)\ln\hat{f_{UV}^{n}}\left(u_{i},v_{i}\right)}{\frac{1}{n}\sum_{i=1}^{n}{\hat{f_{UV}^{n}}}^{\tau }\left(u_{i},v_{i}\right)}\approx \frac{1}{n}\sum_{i=1}^{n}{\hat{f_{UV}^{n}}}^{\tau }\left(u_{i},v_{i}\right)\ln\hat{f_{UV}^{n}}\left(u_{i},v_{i}\right).$$Consequently,
$$\hat{d_{\tau }^{n}}\left(a,b\right)\approx \frac{1}{n}\left[\sum_{i=1}^{n}{\hat{f_{UV}^{n}}}^{\tau }\left(u_{i},v_{i}\right)\ln\hat{f_{UV}^{n}}\left(u_{i},v_{i}\right)-\sum_{i=1}^{n}{\hat{f_{U}^{n}}}^{\tau }\left(u_{i}\right){\hat{f_{V}^{n}}}^{\tau }\left(v_{i}\right)\ln\left(\hat{f_{U}^{n}}\left(u_{i}\right)\hat{f_{V}^{n}}\left(v_{i}\right)\right)\right].$$

Note that this approximation is valid for τ closed to 0, but, for τ=0,
$$d_{0}\left(a,b\right)=\lim_{\tau ↓0}{\tilde{d}}_{\tau }\left(a,b\right)=\frac{1}{n}\sum_{i=1}^{n}\ln\left(\frac{\hat{f_{UV}^{n}}\left(u_{i},v_{i}\right)}{\hat{f_{U}^{n}}\left(u_{i}\right)\hat{f_{V}^{n}}\left(v_{i}\right)}\right)=\frac{1}{n}\left(\sum_{i=1}^{n}\ln\hat{f_{UV}^{n}}\left(u_{i},v_{i}\right)-\sum_{i=1}^{n}\ln\hat{f_{U}^{n}}\left(u_{i}\right)\hat{f_{V}^{n}}\left(v_{i}\right)\right).$$

This implies that $\hat{d_{\tau }^{n}}\left(f_{U}\times f_{V},f_{UV}\right)$ can be seen as a weighted value of the empirical Kullback–Leibler divergence and the weights depend on ${\hat{f_{UV}^{n}}}^{\tau }\left(u_{i},v_{i}\right),{\hat{f_{U}^{n}}}^{\tau }\left(u_{i}\right)$ and ${\hat{f_{V}^{n}}}^{\tau }\left(v_{i}\right).$ Therefore, if the observations $x_{i},y_{i}$ or both are outliers, the corresponding density estimations would decrease, so the corresponding weights would not be considered as important as other data on the estimated distance, thus making Renyi’s pseudodistance more robust to outliers than the Kullback–Leibler.

## 5. Testing to Determine the Number of Pairs

In this section, a dimension reduction algorithm is described for determining the number of significant pairs of canonical vectors: the non-parametric sequential test [41,42].

In the classical approach of CCA, the maximum number of pairs $\left(a_{i},b_{i}\right)$ is determined by the greatest index *j* such that $\left(a_{j},b_{j}\right)$ is the first pair satisfying $\rho (a_{j}^{T}X,b_{j}^{T}Y)=0.$ That is, the CCA should run until the best estimated pair leads to linear independence. A natural extension for the RPCCA formulation is then replicated in the CCA dimension reduction algorithm, but using the RP divergence as a measure of dependence.

Let us denote by $d_{\tau }^{i}$ the maximum value achieved at the *i*-th iteration,
$$d_{\tau }^{i}=\max_{a_{i},b_{i}}d_{\tau }\left(a_{i},b_{i}\right),\;i=1,\ldots ,l=\min\left(q,p\right),$$
such that $a_{i}^{T}\Sigma_{11}a_{i}=$$b_{i}^{T}\Sigma_{22}b_{i}=1$ and $a_{j}^{T}\Sigma_{11}a_{i}=$$b_{j}^{T}\Sigma_{22}b_{i}=0.$ The sequence of maximums is decreasing and lower-bounded by 0, indicating independence between the estimated canonical variables, $d_{\tau }^{1}\geq d_{\tau }^{2}\geq \ldots \geq d_{\tau }^{l}\geq 0.$ Then, a stopping criterion for the maximum number of canonical correlations is naturally determined by the testing problem
$$H_{0}:d_{\tau }^{i}=0\;\mathrm{vs}\;H_{1}:d_{\tau }^{i}>0,\;i=1,\ldots ,l.$$

If $H_{0}$ is not rejected, then all subsequent canonical variables from the *i*-th onward are not significantly related. Otherwise, the relation is significant, and the maximum number of significant canonical correlations is at least i. It is difficult to obtain the exact sample distribution of $d_{\tau }^{i}$, but a non-parametric permutation test can be applied, as proposed in [24], for estimating the *p*-value of the test. Let us explain this procedure with some detail. Suppose there is a relationship between $a_{i}^{T}X$ and $b_{i}^{T}Y$ for some vectors $a_{i},b_{i}$, i.e., $H_{0}$ does not hold. This means that there exists a function *f* such that
$$f\left(a_{i}^{T}X\right)\approx b_{i}^{T}Y.$$

Our procedure will estimate vectors $a_{i}$ and $b_{i}$ and will consider some (near!) vectors ${\hat{a}}_{i}$ and ${\hat{b}}_{i},$ respectively. Consequently, we expect that for the sample $\left(X_{1},Y_{1}\right),\ldots ,\left(X_{n},Y_{n}\right),$ we will obtain
$$f\left({\hat{a}}_{i}^{T}X_{j}\right)\approx {\hat{b}}_{i}^{T}Y_{j},\;j=1,\ldots ,n.$$

This will translate in a large value of $d_{\tau }^{n}\left({\hat{a_{i}}}^{T}X,{\hat{b_{i}}}^{T}Y\right)$ and, consequently, the corresponding estimation $\hat{d_{\tau }^{n}}\left({\hat{a_{i}}}^{T}X,{\hat{b_{i}}}^{T}Y\right).$ Now, if we consider a permutation of the data corresponding to X but maintaining the order for the data corresponding to Y, any possible relationship is destroyed because the data corresponding to $X_{i}$ do not correspond to individual *i* in the sample, so that they have nothing to do with $Y_{i}$. In other words, if we denote the reordered sample for $\left(X_{1},\ldots ,X_{n}\right)$ by $(X_{1}^{*},\ldots ,X_{n}^{*}),$ it follows that for any c,d, then $\hat{d_{\tau }^{n}}\left(c^{T}X^{*},d^{T}Y\right)\approx 0$ showing independence. Consequently, when the procedure looks for some vectors $\hat{a_{i}^{*}},\hat{b_{i}^{*}}$ s.t.
$$\hat{d_{\tau }^{n}}\left({\hat{a_{i}^{*}}}^{T}X^{*},{\hat{b_{i}^{*}}}^{T}Y\right)=\max\hat{d_{\tau }^{n}}\left(a^{T}X^{*},b^{T}Y\right),$$
these values are not expected to model a strong relation (because it does not exist), so that we expect $\hat{d_{\tau }^{n}}\left({\hat{a_{i}^{*}}}^{T}X^{*},{\hat{b_{i}^{*}}}^{T}Y\right)\approx 0.$ Hence, if $H_{0}$ does not hold and a relationship between the canonical variables exists, we expect that for (almost) any permutation
$$\hat{d_{\tau }^{n}}\left({\hat{a_{i}}}^{T}X,{\hat{b_{i}}}^{T}Y\right)>\hat{d_{\tau }^{n}}\left({\hat{a_{i}^{*}}}^{T}X^{*},{\hat{b_{i}^{*}}}^{T}Y\right).$$

On the other hand, if $H_{0}$ holds, then there is independence between $c^{T}X$ and $d^{T}Y$ for any c,d. Consequently, for the best possible estimated vectors $\hat{a_{i}},\hat{b_{i}}$, we will obtain
$$\hat{d_{\tau }^{n}}\left({\hat{a_{i}}}^{T}X,{\hat{b_{i}}}^{T}Y\right)\approx 0.$$

When considering a permutation of the values corresponding to X, independence will arise again, and hence, in this case, we expect
$$\hat{d_{\tau }^{n}}\left({\hat{a_{i}}}^{T}X,{\hat{b_{i}}}^{T}Y\right)\approx \hat{d_{\tau }^{n}}\left({\hat{a_{i}^{*}}}^{T}X^{*},{\hat{b_{i}^{*}}}^{T}Y\right).$$

Of course, the number of possible permutations is n!, and this is not affordable for large values of *n*. Hence, we are going to consider just a subset of randomly chosen permutations. Then, if ${\hat{d}}_{\tau }^{i,w}$ denotes the value of the index corresponding to the *w*-th randomly permuted sample, the estimated *p*-value of the test is given by
$$\frac{1}{R}\sum_{w=1}^{R}I_{\left[{\hat{d}}_{\tau }^{i,w}>{\hat{d}}_{\tau }^{i}\right]},$$
where *R* denotes the number of permutations considered. Yin [12] used R=1000 for a permutation test for ICCA. If the *p*-value is smaller than a certain significance level, then the null hypothesis $d_{\tau }^{i}=0$ should be rejected implying a significant relationship for the *i*-th canonical variables, and the process should be repeated for i+1. Conversely, if the null hypothesis is not rejected, then we should assume that the canonical variables are independent and conclude that there are only *i* estimated canonical variables exhibiting significant relationships. More details about this dimension reduction method can be seen in [24].

## 6. Simulation Study

### 6.1. Computational Methods

Consider $X=(x_{1},\ldots ,x_{n})$ and $Y=(y_{1},\ldots ,y_{n})$ as p×n and q×n matrices with *n* observations of the random variables X and Y, respectively. The estimation of the *i*-th pair of canonical vectors ${\hat{a}}_{i}^{\tau },{\hat{b}}_{i}^{\tau }$ based on the RP with tuning parameter τ is computed through the constrained maximization problem
(11)$$\begin{array}{cc} \; & \left({\hat{a}}_{n}^{\tau },{\hat{b}}_{n}^{\tau }\right)=\arg\max_{a,b}\hat{d_{\tau }^{i}}\left(a,b\right),\\ \; & \;s.t.\;{\left(a_{i}^{\tau }\right)}^{T}{\hat{\Sigma }}_{X}a_{i}^{\tau }=1\;\mathrm{and}\;{\left(b_{i}^{\tau }\right)}^{T}{\hat{\Sigma }}_{Y}b_{n}^{\tau }=1,\\ \; & \;{\left(a_{i}^{\tau }\right)}^{T}{\hat{\Sigma }}_{X}b_{j}^{\tau }=0\;\mathrm{and}\;{\left(b_{i}^{\tau }\right)}^{T}{\hat{\Sigma }}_{Y}b_{j}^{\tau }=0,\;j=1,\ldots ,i-1 \end{array}$$
where ${\hat{\Sigma }}_{X}$ and ${\hat{\Sigma }}_{Y}$ are the empirical estimators of the variance–covariance matrices of the multidimensional random variables X and Y, respectively.

The optimization problem constraints can be simplified by scaling the sample matrices to have zero mean and unit variance as follows:(12)$$\begin{array}{cc} \tilde{X} & ={\hat{\Sigma }}_{X}^{-1/2}\left(X-\bar{X}\right)\\ \tilde{Y} & ={\hat{\Sigma }}_{22}^{-1/2}\left(Y-\bar{Y}\right), \end{array}$$
where $\bar{X}$ and $\bar{Y}$ denote the corresponding sample mean vectors. From Proposition 1, the RPCCA is invariant under such linear transformations, and, consequently, the problem constraints are transformed into
(13)$$\begin{array}{cc} \; & {\left(a_{i}^{\tau }\right)}^{T}a_{i}^{\tau }=1\;\mathrm{and}\;{\left(b_{i}^{\tau }\right)}^{T}b_{n}^{\tau }=1,\\ \; & {\left(a_{i}^{\tau }\right)}^{T}b_{j}^{\tau }=0\;\mathrm{and}\;{\left(b_{i}^{\tau }\right)}^{T}b_{j}^{\tau }=0,\;j=1,\ldots ,i-1. \end{array}$$

For empirical covariance matrices, it may appear as disease degeneration resulting in uninvertible matrices. In those cases, we can skip the scaling transform and apply the estimation algorithm under the original restrictions.

From the transformed canonical vectors, ${\tilde{a}}_{i}$ and ${\tilde{b}}_{i},$ the estimated canonical vectors in the original space can be easily recovered as ${\hat{a}}_{i}={\hat{\Sigma }}_{X}^{-1/2}{\tilde{a}}_{i}$ and ${\hat{b}}_{i}={\hat{\Sigma }}_{Y}^{-1/2}{\tilde{b}}_{i}.$

Here, the constrained optimization is carried out iteratively using the non-linear constrained optimizer *optimize* from the *scipy* package in *Python*, which implements a Sequential Quadratic Programming (SQP) method. The source code for the implementation is publicly available on https://github.com/MariaJaenada/Robust-Canonical-Correlations (Github) (accessed on 29 January 2023).

### 6.2. Monte Carlo Simulation

We empirically examine the robustness of the RPCCA method through a Monte Carlo simulation. We consider a pair of random vectors, $X=(X_{1},\ldots ,X_{8})$ and $Y=(Y_{1},Y_{2},Y_{3}),$ whose components satisfy a linear and a non-linear relationship of the form:(14)$$Y_{1}={\left(2X_{1}+X_{2}+X_{3}\right)}^{2}\;\mathrm{and}\;\;Y_{2}=X_{2}-X_{3}.$$

The rest of the variables are independent and they are defined as follows: $X_{1},X_{2},X_{6},X_{7}$ and $X_{8}$. They are standard normal variables. $X_{3}$ comes from a chi-square distribution with 7 degrees of freedom, $X_{4}$ follows a *t*-Student distribution with 5 degrees of freedom and $X_{5}$ comes from a Fisher–Snedecor distribution with 3 and 12 degrees of freedom, respectively. Finally, $Y_{3}$ comes from a *t*-Student with 9 degrees of freedom.

The true underlying canonical vectors are then $a_{1}=\left(0,\;1,-1,\;0,\;0,\;0,\;0,\;0\right),$$b_{1}=\left(0,\;1,\;0\right)$ and $a_{2}=\left(2,\;1,\;1,\;0,\;0,\;0,\;0,\;0\right),$$b_{2}=\left(1,\;0,\;0\right).$ Note that they are orthogonal, and so are the related variables $Y_{1}$ and $Y_{2}.$ Although in the procedure we compute unit-norm vectors, we have considered the description vectors with natural coefficients as they look easier to understand. We named the first canonical vector $a_{1}$ because we empirically detected that the linear relationship is first captured.

We generate a random sample of the pairs X and Y of size n=100, and we estimate the pairs of canonical vectors ${\hat{a}}_{i}$ and ${\hat{b}}_{i},$i=1,2, such that the random variables $U_{i}=a_{i}^{T}X$ and $V_{i}=b_{i}^{T}Y$ are functionally interrelated. To examine the performance of the RPCCA method under contamination, we randomly switch the functional relationships in Equation (Equation 14) for an ε% of the observations, with ε=5,10,15 and 20 denoting the contamination proportion. That is, for a random ε% of the observations, the values of $Y_{1}$ and $Y_{2}$ are exchanged, generating orthogonal outliers; the functions defining the $Y_{2}$ and $Y_{1}$ are orthogonal to each other. Therefore, this contamination will worsen both relationships at the same time in orthogonal directions. We repeat the simulations over R=500 replications and compute averages of the following performance measures: We quantify the accuracy of the estimates with the absolute correlations between the estimated and true canonical variables, $|\rho \left(a_{i},{\hat{a}}_{i}\right)|=|\rho \left(a_{i}^{T}X,{\hat{a}}_{i}^{T}X\right)|$ and $|\rho \left(b_{i},{\hat{b}}_{i}\right)|=|\rho \left(b_{i}^{T}Y,{\hat{b}}_{i}^{T}Y\right)|.$ Additionally, to evaluate the robustness of the method, we compute the $L_{2}$-norm between the canonical vectors fitted under uncontaminated and contaminated data, $\hat{a}$ and ${\hat{a}}^{c},$
$$L_{2}\left(\hat{a},{\hat{a}}^{c}\right)=\left|\right|\hat{a}-{\hat{a}}^{c}{\left|\right|}_{2}$$
as well as the projection of ${\hat{a}}^{c}$ into the orthogonal subspace spanned by the uncontaminated estimate, $\hat{a},$
$$P_{2}\left(\hat{a},{\hat{a}}^{c}\right)=\left|\right|\left(I-\hat{a}{\hat{a}}^{T}\right){\hat{a}}_{c}^{T}{\left|\right|}_{2}.$$

The distance measures $L_{2}\left(\hat{a},{\hat{a}}^{c}\right)$ and $P_{2}\left(\hat{a},{\hat{a}}^{c}\right)$ are smaller the more stable the estimate is, implying that the estimates are not largely affected by the contamination; hence the corresponding method is more robust. Summarizing, the correlations between true and estimated canonical variables ρ(·,·) aim to represent the accuracy of the method, whereas the distance measures between estimated canonical vectors for pure data and for contaminated data, $L_{2}$ and $P_{2}$, aim to represent the robustness of the method.

Table 1 and Table 2 present all performance measures for the RPCCA method over a grid of tuning parameters ranging from 0 (corresponding to ICCA) to 0.8. All methods perform suitably well in terms of accuracy, achieving high absolute correlations between true and estimated canonical variables, even under contaminated scenarios. However, the linear relationship in the first component is captured worse by the ICCA in the presence of contamination, as shown by the lower absolute correlations between the canonical variables, $\rho ({\hat{a}}_{1}^{c},{\hat{b}}_{1}^{c}).$ Moreover, the RPCCA method with positive values of the tuning parameter produces more stable estimations of the canonical vectors, having smaller $P_{2}$ and $L_{2}$ distances between the uncontaminated and contaminated estimated canonical vectors in both components, $\left(\hat{b_{1}},{\hat{b}}_{1}^{c}\right)$ and $(\hat{b_{2}},{\hat{b}}_{2}^{c}),$ thus demonstrating the advantage in terms of robustness. Although the differences in performance are not impressive, the gain in robustness with very little loss of accuracy with respect to the ICCA makes the RPCCA very attractive.

On the other hand, if the underlying relationship is easily identified, the proposed robust RPCCA performs as good as the ICCA under pure data and outperforms the ICCA in the presence of contamination (Table 1). However, for τ>0, the loss in accuracy in the relationship identification under pure data would be unavoidable (although not very significant); hence, the tuning parameter should be chosen sufficiently close to zero (from the literature, less than 1) to provide an adequate compromise between efficiency loss and robustness gain. Moderate values of the tuning parameter, around 0.3, offer the best compromise producing canonical estimators that are robust against data contamination with a small loss of efficiency with respect to the ICCA in the absence of contamination.

### 6.3. Real Data Application

We finally illustrate the applicability of our method with real-life data on the heredity of head shape in men. For such a purpose, we use a well-known dataset from Frets [43] that collects the head length and head breadth for the first and second sons for n=25 families. Then, the first and second set of variables, X and Y, respectively, have the dimension 2 and represent the head length and head breadth of the corresponding son. From the dataset, we want to analyze whether there is a relationship between the head shape among male offspring. The data have been widely used in the literature, and Mardia et al. [2] and Yin [12] analyzed the canonical correlations between the first and second sons’ head shapes using CCA and ICCA, respectively. In their analyses, they found one significant pair of canonical variables with a strong linear relationship. Figure 2 shows the plots of the first (left) and second (right) pair of canonical variables for the head data estimated by RPCCA with τ=0 (top) and τ=0.5 (bottom). As shown, both methods coincide on the first pair of canonical variables (estimate the same observations for the first pair of canonical variables), $x_{1}$ and $y_{1},$ having linear correlation coefficients of ρ=78.67% (τ=0) and ρ=76.86% (τ=0.5) as illustrated on the corresponding plots. For the second pair of canonical variables, none of the methods find any clear functional relationship between linear combinations of the variables, and the two procedures considered estimate very different canonical variables without a clear functional relationship between them (as shown in Figure 2). Thus, we also conclude that there is only one pair of canonical variables.

Additionally, to illustrate the advantage of our method in terms of robustness (with a small loss of deficiency), we contaminate a single observation (obs. 24) in both vector variables, generating an outlying observation. Then, we apply RPCCA at τ=0 (corresponding to ICCA) and τ=0.5 with the uncontaminated and the contaminated data. Table 3 presents $P_{2}$ and $N_{2}$ distances between the first pair of canonical vectors (identifying the linear relationship) estimated under uncontaminated and contaminated data, with only one outlying observation. Because the sample size is small, an outlying observation heavily influences the ICCA estimation, whereas the RPCCA method with τ=0.5 shows a great stability in the canonical vector estimation. These results illustrate the advantage of the RPCCA in real-life applications, producing robust estimates of the canonical variables with a small loss of efficiency with respect to the ICCA estimation in the absence of data contamination.

## 7. Conclusions

We have presented a robust generalization of the ICCA based on RP for identifying linear and non-linear relationships between two sets of variables. We have derived sample versions for estimating the canonical vectors in practice, and we have demonstrated the consistency of such estimators. Further, the robustness advantage of the RPCCA has been examined theoretically and empirically, concluding that the proposed RPCCA offers an appealing alternative to ICCA, competitive in terms of estimation accuracy and more robust against data contamination. The method manages to detect hidden functional relationships between linear combinations of the variables and suitably approximates the true underlying relationships, even under contaminated scenarios. Moreover, a permutation test for determining the number of significant pairs of canonical vectors is presented. Since the RPCCA is a parametric family, a data-driven algorithm for determining optimal values of the tuning parameter is a worthwhile pursuit for future research. Also, the methodology presented here can be extended in future works for identifying relationships between more than two sets of variables. The idea is to consider not only two random vectors but *k* random vectors as considered in [24] and to look for the linear combinations in all of them so that the RP between the marginal distributions and the whole distribution is as large as possible.

## Figures and Tables

**Figure 1 entropy-25-00713-f001:**
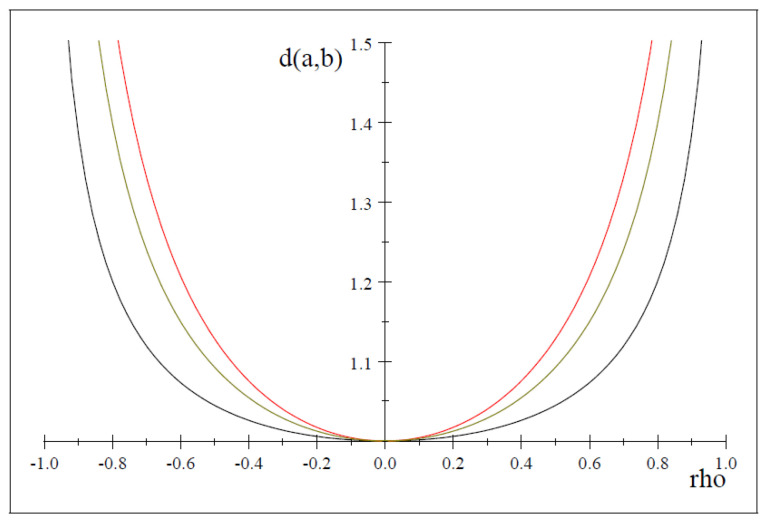
$f_{\tau }\left(\rho \right)$ for different values of τ.τ=0.1 (red), τ=0.3 (green) and τ=0.9 (black).

**Figure 2 entropy-25-00713-f002:**
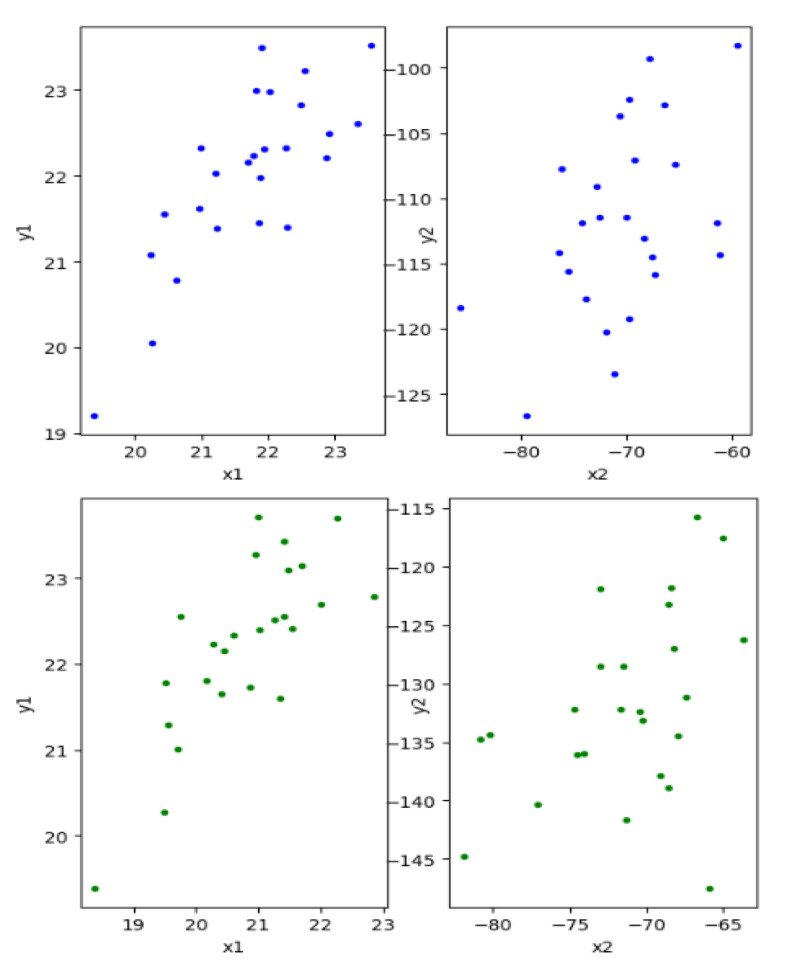
Pairs of canonical variables obtained from RPCCA with τ=0 (**top**) and τ=0.5 (**bottom**) for the head dataset.

**Table 1 entropy-25-00713-t001:** RPCCA error measures for the first canonical vector under different values of the tuning parameter τ.

τ	0	0.1	0.2	0.3	0.4	0.5	0.6	0.7	0.8
Pure data									
$\rho ({\hat{a}}_{1},{\hat{b}}_{1})$	0.92878	0.96111	0.95505	0.97044	0.97099	0.98082	0.97090	0.98256	0.98606
$\rho (a_{1},{\hat{a}}_{1})$	0.99908	0.99907	0.99893	0.99877	0.99856	0.99831	0.99799	0.99767	0.99728
$\rho (b_{1},{\hat{b}}_{1})$	0.99946	0.99951	0.99933	0.99938	0.99936	0.99931	0.99906	0.99916	0.99899
5% contamination									
$\rho ({\hat{a}}_{1}^{c},{\hat{b}}_{1}^{c})$	0.65796	0.68689	0.71486	0.75556	0.76384	0.80646	0.80876	0.81990	0.81201
$\rho (a_{1},{\hat{a}}_{1}^{c})$	0.99801	0.99815	0.99805	0.99786	0.99749	0.99645	0.99353	0.98368	0.96927
$\rho (b_{1},{\hat{b}}_{1}^{c})$	0.99548	0.99571	0.99531	0.99538	0.99461	0.99345	0.99084	0.97936	0.96222
$P_{2}\left(\hat{a_{1}},{\hat{a}}_{1}^{c}\right)$	0.41125	0.38246	0.36111	0.31247	0.30856	0.25109	0.25272	0.22822	0.23050
$P_{2}\left(\hat{b_{1}},{\hat{b}}_{1}^{c}\right)$	0.35685	0.32477	0.31442	0.27277	0.26949	0.22345	0.22962	0.20914	0.21654
$L_{2}\left(\hat{a_{1}},{\hat{a}}_{1}^{c}\right)$	0.56677	0.52712	0.49669	0.42880	0.42197	0.33991	0.34109	0.30249	0.30241
$L_{2}\left(\hat{b_{1}},{\hat{b}}_{1}^{c}\right)$	0.41496	0.37628	0.36696	0.31681	0.31561	0.26241	0.27298	0.24999	0.26391
10% contamination									
$\rho ({\hat{a}}_{1}^{c},{\hat{b}}_{1}^{c})$	0.41963	0.43878	0.46604	0.49193	0.53443	0.56155	0.57944	0.59613	0.60565
$\rho (a_{1},{\hat{a}}_{1}^{c})$	0.99698	0.99714	0.99712	0.99686	0.99563	0.99225	0.98450	0.96631	0.95789
$\rho (b_{1},{\hat{b}}_{1}^{c})$	0.99054	0.99018	0.98974	0.98968	0.98719	0.98401	0.97274	0.95167	0.93961
$P_{2}\left(\hat{a_{1}},{\hat{a}}_{1}^{c}\right)$	0.68137	0.65989	0.63978	0.60585	0.57014	0.53623	0.51301	0.48214	0.47437
$P_{2}\left(\hat{b_{1}},{\hat{b}}_{1}^{c}\right)$	0.56499	0.54370	0.53006	0.51765	0.48283	0.46007	0.44608	0.43238	0.43151
$L_{2}\left(\hat{a_{1}},{\hat{a}}_{1}^{c}\right)$	0.94237	0.91248	0.88343	0.83560	0.78292	0.73427	0.69877	0.64896	0.63289
$L_{2}\left(\hat{b_{1}},{\hat{b}}_{1}^{c}\right)$	0.63783	0.61685	0.60406	0.59298	0.55556	0.53199	0.52065	0.51211	0.51843
15% contamination									
$\rho ({\hat{a}}_{1}^{c},{\hat{b}}_{1}^{c})$	0.26726	0.29776	0.32650	0.34987	0.40472	0.42547	0.43099	0.43524	0.45246
$\rho ({\hat{a}}_{1},{\hat{a}}_{1}^{c})$	0.99662	0.99682	0.99670	0.99631	0.99405	0.98763	0.97539	0.95782	0.93472
$\rho ({\hat{b}}_{1},{\hat{b}}_{1}^{c})$	0.98740	0.98702	0.98627	0.98541	0.98364	0.97541	0.95686	0.93536	0.91078
$P_{2}\left(\hat{a_{1}},{\hat{a}}_{1}^{c}\right)$	0.83673	0.81093	0.78650	0.75948	0.70320	0.68752	0.68190	0.67188	0.64529
$P_{2}\left(\hat{b_{1}},{\hat{b}}_{1}^{c}\right)$	0.67519	0.65622	0.64107	0.62355	0.58194	0.57677	0.58054	0.58166	0.57108
$L_{2}\left(\hat{a_{1}},{\hat{a}}_{1}^{c}\right)$	1.16074	1.12367	1.08848	1.04925	0.96766	0.94157	0.92937	0.90983	0.86243
$L_{2}\left(\hat{b_{1}},{\hat{b}}_{1}^{c}\right)$	0.75424	0.73561	0.72109	0.70581	0.66064	0.66160	0.67484	0.68431	0.68025
20% contamination									
$\rho ({\hat{a}}_{1}^{c},{\hat{b}}_{1}^{c})$	0.19852	0.21736	0.22524	0.23800	0.27663	0.28001	0.30514	0.32119	0.31858
$\rho ({\hat{a}}_{1},{\hat{a}}_{1}^{c})$	0.99661	0.99683	0.99652	0.99593	0.99429	0.99003	0.96500	0.94362	0.90237
$\rho ({\hat{b}}_{1},{\hat{b}}_{1}^{c})$	0.98527	0.98503	0.98336	0.98251	0.97950	0.97282	0.94210	0.91641	0.86912
$P_{2}\left(\hat{a_{1}},{\hat{a}}_{1}^{c}\right)$	0.90198	0.90524	0.89112	0.89756	0.85384	0.85092	0.82023	0.80367	0.79283
$P_{2}\left(\hat{b_{1}},{\hat{b}}_{1}^{c}\right)$	0.72434	0.71723	0.71807	0.72166	0.69817	0.70049	0.69510	0.69435	0.70935
$L_{2}\left(\hat{a_{1}},{\hat{a}}_{1}^{c}\right)$	0.45211	0.45326	0.44686	0.45094	0.42851	0.42746	0.41497	0.41101	0.41076
$L_{2}\left(\hat{b_{1}},{\hat{b}}_{1}^{c}\right)$	0.61339	0.61723	0.60407	0.60759	0.59650	0.59591	0.58441	0.59536	0.59885

**Table 2 entropy-25-00713-t002:** RPCCA error measures for the second canonical vector under different values of the tuning parameter τ.

τ	0	0.1	0.2	0.3	0.4	0.5	0.6	0.7	0.8
Pure data									
$\rho ({\hat{a}}_{2},{\hat{b}}_{2})$	0.22829	0.19656	0.20033	0.18556	0.18386	0.17334	0.17650	0.16407	0.15426
$\rho (a_{2},{\hat{a}}_{2})$	0.99687	0.99704	0.99514	0.99387	0.98976	0.96917	0.94815	0.93493	0.88471
$\rho (b_{2},{\hat{b}}_{2})$	0.99464	0.99490	0.99122	0.98980	0.98375	0.96336	0.93706	0.91885	0.85931
5% contamination									
$\rho ({\hat{a}}_{2}^{c},{\hat{b}}_{2}^{c})$	0.30303	0.29521	0.27510	0.24815	0.24522	0.21903	0.22096	0.19770	0.20928
$\rho (a_{2},{\hat{a}}_{2}^{c})$	0.91167	0.92137	0.91977	0.91112	0.91556	0.90693	0.88127	0.84600	0.82650
$\rho (b_{2},{\hat{b}}_{2}^{c})$	0.88049	0.89166	0.88662	0.88904	0.89241	0.88801	0.87192	0.83197	0.80588
$P_{2}\left(\hat{a_{2}},{\hat{a}}_{2}^{c}\right)$	0.41815	0.38347	0.36114	0.32231	0.31958	0.29261	0.33041	0.32611	0.36974
$P_{2}\left(\hat{b_{2}},{\hat{b}}_{2}^{c}\right)$	0.44509	0.41272	0.40340	0.36543	0.36662	0.33627	0.36107	0.34812	0.36182
$L_{2}\left(\hat{a_{2}},{\hat{a}}_{2}^{c}\right)$	0.56221	0.51686	0.48609	0.43221	0.42597	0.38524	0.42792	0.41586	0.46555
$L_{2}\left(\hat{b_{2}},{\hat{b}}_{2}^{c}\right)$	0.54952	0.50631	0.48727	0.43321	0.43065	0.38858	0.41934	0.39996	0.41901
10% contamination									
$\rho ({\hat{a}}_{2}^{c},{\hat{b}}_{2}^{c})$	0.31085	0.29651	0.28563	0.28324	0.25836	0.24660	0.25507	0.23689	0.22360
$\rho (a_{2},{\hat{a}}_{2}^{c})$	0.76633	0.76784	0.77030	0.77828	0.75962	0.77061	0.78303	0.75727	0.71201
$\rho (b_{2},{\hat{a}}_{2}^{c})$	0.77035	0.75135	0.75242	0.75593	0.74394	0.75508	0.74693	0.74538	0.70453
$P_{2}\left(\hat{a_{2}},{\hat{a}}_{2}^{c}\right)$	0.69844	0.67264	0.64760	0.62480	0.58957	0.56764	0.57146	0.56234	0.60946
$P_{2}\left(\hat{b_{2}},{\hat{b}}_{2}^{c}\right)$	0.70333	0.68232	0.65851	0.63732	0.60797	0.57351	0.55445	0.54736	0.53981
$L_{2}\left(\hat{a_{2}},{\hat{a}}_{2}^{c}\right)$	0.94522	0.91180	0.87884	0.84254	0.79453	0.76052	0.75866	0.73627	0.79154
$L_{2}\left(\hat{b_{2}},{\hat{b}}_{2}^{c}\right)$	0.87344	0.84137	0.80061	0.76733	0.72179	0.67148	0.64907	0.64158	0.63347
15% contamination									
$\rho ({\hat{a}}_{2}^{c},{\hat{b}}_{2}^{c})$	0.29041	0.26604	0.25341	0.24867	0.22370	0.22678	0.22668	0.21698	0.20791
$\rho ({\hat{a}}_{2},{\hat{b}}_{2}^{c})$	0.62670	0.62511	0.62775	0.63798	0.63931	0.67347	0.66306	0.63881	0.61437
$\rho ({\hat{b}}_{2},{\hat{b}}_{2}^{c})$	0.67113	0.65281	0.63298	0.64345	0.64183	0.66804	0.66294	0.66655	0.64393
$P_{2}\left(\hat{a_{2}},{\hat{a}}_{2}^{c}\right)$	0.86840	0.83480	0.80722	0.78144	0.73384	0.71580	0.71970	0.73208	0.73667
$P_{2}\left(\hat{b_{2}},{\hat{b}}_{2}^{c}\right)$	0.83973	0.81076	0.78283	0.74695	0.69787	0.68081	0.65889	0.63941	0.61576
$L_{2}\left(\hat{a_{2}},{\hat{a}}_{2}^{c}\right)$	1.17901	1.13476	1.09703	1.06197	0.99294	0.96654	0.96359	0.97065	0.96509
$L_{2}\left(\hat{b_{2}},{\hat{b}}_{2}^{c}\right)$	1.04721	1.00181	0.95442	0.89936	0.83020	0.80219	0.77413	0.74676	0.72249
20% contamination									
$\rho ({\hat{a}}_{2}^{c},{\hat{b}}_{2}^{c})$	0.23981	0.23457	0.22347	0.22122	0.21361	0.20732	0.20959	0.19139	0.20603
$\rho ({\hat{a}}_{2},{\hat{b}}_{2}^{c})$	0.51483	0.51794	0.52276	0.53520	0.54956	0.53813	0.56693	0.54726	0.56153
$\rho ({\hat{b}}_{2},{\hat{b}}_{2}^{c})$	0.62369	0.59115	0.56066	0.56134	0.58636	0.58110	0.60681	0.61363	0.63613
$P_{2}\left(\hat{a_{2}},{\hat{a}}_{2}^{c}\right)$	0.94170	0.92377	0.91740	0.90923	0.86834	0.86758	0.83873	0.82330	0.83708
$P_{2}\left(\hat{b_{2}},{\hat{b}}_{2}^{c}\right)$	0.89672	0.87587	0.85688	0.82781	0.78143	0.75837	0.72490	0.70158	0.69711
$L_{2}\left(\hat{a_{2}},{\hat{a}}_{2}^{c}\right)$	0.47541	0.46413	0.46168	0.45820	0.43798	0.43988	0.43203	0.42330	0.43539
$L_{2}\left(\hat{b_{2}},{\hat{b}}_{2}^{c}\right)$	0.76644	0.73624	0.72641	0.69977	0.67411	0.63579	0.61011	0.61020	0.59094

**Table 3 entropy-25-00713-t003:** $P_{2}$ and $N_{2}$ distances between the estimated canonical vectors under uncontaminated and contaminated data.

τ	0	0.5
$P_{2}\left(\hat{a_{1}},{\hat{a}}_{1}^{c}\right)$	0.222	0.064
$P_{2}\left(\hat{b_{1}},{\hat{b}}_{1}^{c}\right)$	0.131	0.059
$L_{2}\left(\hat{a_{1}},{\hat{a}}_{1}^{c}\right)$	0.224	0.064
$L_{2}\left(\hat{b_{1}},{\hat{b}}_{1}^{c}\right)$	1.995	0.059

## Data Availability

Not applicable.

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
