# Peer review of "An Approach to Canonical Correlation Analysis Based on Rényi’s Pseudodistances"

_entropy, 2023, doi:10.3390/e25050713_

Round 1
Reviewer 1 Report
See the attached report.

Author Response
Thank you very much for your report. In the new version of the paper we have tried to cope with your suggestions. See the attached file for details. Changes in the manuscript are highlighted in red.

Reviewer 2 Report
Attached document

Author Response

(The authors gave the same response as above.)

Reviewer 3 Report
I cannot find problem formulation. Statement “CCA determines pairwise linear relationships between two sets of random vectors, namely X and Y” means nothing. What is nature of random vectors X and Y? What is type of relationship? What real life problem can be reformulated as problem for CCA? Later authors state that ICCA can define existence of nonlinear relationships. It is well known that linear relationship is unique and can be easily identified. From the other side, nonlinear relationship is not unique and fact of existence of such relationship cannot help to find it. As a result, content of paper is completely unclear because there is no any problem to solve. I should state again that CCA is tool to solve some problem but not a real life problem.
Description of example in the first paragraph of introduction is incomplete or inaccurate. If Y_1=X_1^2+Z then cov(Y_1,X_1) = cov(X_1^2+Z,X_1)=cov(X_1^2,X_1)+ cov(Z,X_1)= cov(X_1^2,X_1). According to X~N(0,I) we have var(X_1)=1, mean(X_1)=1. This means that mean(X_1^2)=1. This means that cov(X_1^2,X_1)=mean((X_1^2-1)X_1)= mean(X_1^3)-mean(X_1)= mean(X_1^3). Conclusion that mean(X_1^3)=0 is unclear. Moreover, it is not true for most cases. The simplest counterexample is the one positive X_11=a and n negative cases X_12-X_1(n+1)=-a/n. In this case we have mean(X_1)=(a-na/n)/(n+1)=0 and mean(X_1^3)=(a^3-n(a/n)^3)/(n+1)=(a^3-a^3/n^2)/(n+1)>0. From the fact that X_1 and X_2 are uncorrelated cannot be derived fact that X_1^2 and X_2 are also uncorrelated. Counterexample also can be easily created. This means that statement that cov(X,Y) zero matrix is wrong.
Line 17. Statement “Consequently, classical CCA cannot detect the pair of linear combinations of the variables functionally related.” Is incorrect. It can detect linear functional dependence and cannot detect non-linear ones.
Lines 18-21. It is not clear dependency of normal distribution and independence of random variables. Notion of independence of random variable does not have any relation to type of distribution.
Line 22. Statement “It is not surprising that CCA fails here” is unclear because notion “here” is not defined.
Line 22. Statement “but the true relation underlying is quadratic” is wrong since context is not described.
Line 46. Statement “q=min(q,p)” is slightly strange. I think it can be rewritten in better way. For example simple q<p.
Line 94. Statement that “the canonical variables have unit variance” is something strange. This property is not caused by fitting parameters. It can be caused by algorithm of search but this algorithm is not described.
Paper should be described for reader outside the authors team. It is absolutely necessary to describe real life problem, reasons to consider non-linear relationships, proposed algorithm, etc.
Technical comments
Line 15 “pair-wise” must be “pairwise”.
Author Response

(The authors gave the same response as above.)

Reviewer 4 Report
I appreciated reading this paper. This paper is well written and organized.
I have only some minor comments. Particularly, I suggest that authors emphasize limitations and further research streams in the conclusion section.
I also suggest to use a sample with real data to test the performance and robustness of the proposed technique.
Author Response

(The authors gave the same response as above.)

Round 2
Reviewer 1 Report
The authors answered to all the points mentioned in report. In my opinion, the paper can be accepted for publication in this form.
Author Response
We want to thank you very much for your suggestions that have improved the final version of the paper.
Reviewer 2 Report
Typographical errors
Page 3 of 20 (line 99). It says “An example with real data is studied in Section 7” and should say “An example with real data is studied in Section 6”
Page 15 of 20 (line 312). It says “from Frets [Frets]” and should say “from Frets [10]”
Page 19 of 20 (line 397). It says “Frets G.P. Frets, Heredity….” and should say “[10] G.P. Frets, Heredity….”
Author Response
Thank you very much for your reports. In this new version, we have corrected the typos.
Reviewer 3 Report
I cannot find problem formulation. Statement “CCA determines pairwise linear relationships between two sets of random vectors, namely X and Y” means nothing. What is nature of random vectors X and Y? What is type of relationship? What real life problem can be reformulated as problem for CCA? Later authors state that ICCA can define existence of nonlinear relationships. It is well known that linear relationship is unique and can be easily identified. From the other side, nonlinear relationship is not unique and fact of existence of such relationship cannot help to find it. As a result, content of paper is completely unclear because there is no any problem to solve. I should state again that CCA is tool to solve some problem but not a real life problem.
Description of example in the first paragraph of introduction is incomplete or inaccurate. If Y_1=X_1^2+Z then cov(Y_1,X_1) = cov(X_1^2+Z,X_1)=cov(X_1^2,X_1)+ cov(Z,X_1)= cov(X_1^2,X_1). According to X~N(0,I) we have var(X_1)=1, mean(X_1)=1. This means that mean(X_1^2)=1. This means that cov(X_1^2,X_1)=mean((X_1^2-1)X_1)= mean(X_1^3)-mean(X_1)= mean(X_1^3). Conclusion that mean(X_1^3)=0 is unclear. Moreover, it is not true for most cases. The simplest counterexample is the one positive X_11=a and n negative cases X_12-X_1(n+1)=-a/n. In this case we have mean(X_1)=(a-na/n)/(n+1)=0 and mean(X_1^3)=(a^3-n(a/n)^3)/(n+1)=(a^3-a^3/n^2)/(n+1)>0. From the fact that X_1 and X_2 are uncorrelated cannot be derived fact that X_1^2 and X_2 are also uncorrelated. Counterexample also can be easily created. This means that statement that cov(X,Y) zero matrix is wrong.
Line 17. Statement “Consequently, classical CCA cannot detect the pair of linear combinations of the variables functionally related.” Is incorrect. It can detect linear functional dependence and cannot detect non-linear ones.
Lines 18-21. It is not clear dependency of normal distribution and independence of random variables. Notion of independence of random variable does not have any relation to type of distribution.
Line 22. Statement “It is not surprising that CCA fails here” is unclear because notion “here” is not defined.
Line 22. Statement “but the true relation underlying is quadratic” is wrong since context is not described.
Line 46. Statement “q=min(q,p)” is slightly strange. I think it can be rewritten in better way. For example simple q<p.
Line 94. Statement that “the canonical variables have unit variance” is something strange. This property is not caused by fitting parameters. It can be caused by algorithm of search but this algorithm is not described.
Paper should be described for reader outside the authors team. It is absolutely necessary to describe real life problem, reasons to consider non-linear relationships, proposed algorithm, etc.
Technical comments
Line 15 “pair-wise” must be “pairwise”.
I should emphasise that entropy is journal for wide range of reader. This means that each problem MUST be clearly formulated. Authors do not formulate problem under consideration. They simple refer “Besides, in [36] and [13] several real data situations where this philosophy is applied can be found”. I think that at least one real life problem MUST be described to illustrate what is nature of two sets of random vectors and why I cannot join them to one set. This also necessary to illustrate difference between linear and nonlinear relationships and why it is important to use nonlinear while linear more robust and cheap.
In current form paper is not scientific paper because there is no problem to solve. Play with abstract sets of random vectors is meaningless because even for independent sets of vectors we can find some correlation because of finite size of sample.
Authors prepared some explanation for reviewer but forgot include it to text. These explanations contains several highly inaccurate statements. Since these explanations is not part of text I will not consider them. The only recommendation: read about PCA to avoid multicollinearity problem. It is essentially cheaper, faster and robust that any CCA.
Lines 27-30. It is good idea to add fact that for normally distributed random variables uncorrelated variables are independent. I understand that it is trivial but I am sure that such recall helps to reader.
Technical suggestion: why you do not use \Sigma_X and \Sigms_Y instead of \Sigma_11 and \Sigma_22? It will be simple but it is not mandatory.
Lines 66-67. Statement that “the Kullback-Leibler divergence association measure is quite sensitive to outlying observations” should be demonstrated or contains reference. This should be clearly stated what type of outliers you mean because of outliers is defined only for some model (relationship, distribution, etc.). This question is especially important for real life data where we do not know distributions and do not have any models before study. Since Kullback-Leibler divergence is particular case of Rényi’s Pseudodistance then Rényi’s Pseudodistance is also sensitive to outliers. If no it should be demonstrated for which valies of tau.
In lines 67-68 and lines after 110 you wrote “the canonical variables have unit variance” and present equality for canonical vectors. It is good idea to demonstrate why presented restriction for canonical vectors require mentioned properties of canonical variables
Remark 1 is meaningless without description of algorithm of canonical vectors search (phrase “Then, RPCCA procedure finds pairwise canonical vectors a_i \in R^q and b_i in R^p, where i <= q <= p such that” cannot be considered as algorithm description. It is also important that formula after formula (2) does not have any restriction which provide linear independence of found vectors. Moreover, all pairs must be the same according to presented formula). It also necessary to distinguish canonical variables and canonical vectors: you wrote about variables and presented formulae for vectors. It is also unclear which maximum mentioned in Remark.
Example before line 117 looks like incomplete: can you calculate value of d_tau?
Line 123. Normally term nonzero should be used instead of non-null.
Footnote 1. I cannot understand meaning of matrix C multiplied by random variable U. Result will be matrix and I am not sure that RP distance defined for matrices instead of variables (scalars).
It is not good idea to use letter d in Proposition 1 because in proof we can observe fragment cddudv where the first d is coefficient and the second and third d is part of differential.
Proof contains error/typo: F_{UV}^\tau(u,v) must be F_{UV}(u,v).
Covariance matrix in proof of preposition 2. It will e very reasonable to present it in form
\Sigma_X \Sigma_{XY}
\Sigma_{XY}^{-1} \Sigma_Y
In this case it is clear that two rectangular matrices are transpose to each other.
It can be good idea to use different brackets in formula after line 136.
Lines 221-226. Since you can estimate “the Kullback-Leibler divergence” in the same way, does it mean that sensitivity of “the Kullback-Leibler divergence” related to method of estimation but not to “the Kullback-Leibler divergence” itself? Since you mentioned this property please present explanation.
Page 13. “sequence of maximums is strictly increasing” must be decreasing.
Page 13. After reference [14] text is at least inaccurate but can be wrong. There are several questions. What type of permutation is used? Do you mean permutation of part of vector X with respect to other part of vector X? Do you mean permutation of vector X with respect to vector Y? Why you want to calculate distance for fixed i and tau? Problem of selection of tau was not considered. But w can fix it before study from some additional reasons (if can). Why you consider fixed index instead of all indices? Your statement that for independent X and Y permutation “should preserve such independence” seems incorrect. It relates to one problem which was not discussed in paper. If I correctly understand, then it is assumed that one observation of X and Y are related to one object. Otherwise, any discussion of correlation is meaningless. If my assumption is correct then permutation can change “dependence” of X and Y. For simplicity, let us consider one dimensional X and Y: X=(-1, -1, 1, 1) and Y=(-1, 1, 1, -1). In this case we have dot product (covariance) (X,Y) = 1-1+1-1=0 and it looks like independent. After permutation we have X*=(-1, 1, 1, -1) and dot product is (X*,Y)=1+1+1+1=4>0. We observed the strong as possible independence (in this case standard deviation of X and Y is 2 and correlation coefficient corr(X*,Y)=1). Please describe permutation test in detail with all assumptions of p-value estimation.
Line 237. Statement that RP in permutation is less than in original sample is unproved and in general case is wrong. Moreover, this statement cause mianiglessness of the method of p-value estimation.
Line 242. Statement “and the process should be repeated for i + 1” seems strange. Why we cannot produce all calculations once and then select maximal i for which p-value become significant?
Line 256. Very important transformation is unnumbered. I should say that for empirical covariance matrices it is very usual disease – degeneration. This means that rank of matrix is less then dimension and inversion of matrix is impossible. Authors ignored this problem.
Line 260. “under the original matrices” must be “in original space”.
Lines 261-263. If you describe used software please add reference.
Line 269. “The previous canonical vectors”??? Previous to what? Do you mean “previous to this study”?
Lines 266-271. What is distribution of X_i? What is Y_3? Why do you think that “the linear relationship is first captured”? May be you firstly describe dataset (lines 272-279) and only then start description of result?
Line 269. Why your vectors a_1 and a_2 do not have unit length in any norm? What are corresponding vectors b_1 and b_2?
Line 285. You repeated calculation 500 times. What you calculate as a result? Do you calculate mean value for each indicator (rho, L2, P2)? Do you calculate average canonical vector and then calculate rho, L2 and P2 for this average vector? What is variance of calculated canonical vectors? It is clear, that this variance is the best measure of robustness: the smaller variance (may be not variance but coefficient of variation) the more robust method. I should stress that robustness corresponds to less variance but not accuracy I any sense.
Line 287. You wrote “The distance measures L_2(a, a_c) and P_2(a, a_c) are small if the estimation is stable”. What value are small? It is necessary for reading of Tables 1 and 2. For example, in table 1 I can see L_2(a_1, a_1^c) up to 1.16. Is it small? Is it big? One more question: why in line 287 you use a_c but in tables you use a^c?
Line 300. Authors forgot to say that RPCCA is only slightly more robust than ICCA.
Lines 303-304. What do you mean “the efficiency loss”? Do you mean loss of accuracy in the nonlinear relationship identification? If I correctly understand, then this “the efficiency loss” means non-robustness of method. According to table 2, for raw data, correlation between pure canonical variable and estimated canonical variable decrease with increasing of tau.
I also should say that comparison with “true” canonical vectors may be not the best approach. To estimate robustness of method it is necessary to compare canonical variable estimated for pure data and for contaminated data. This comparison will estimate stability (robustness) of estimation ignore accuracy of estimation. Now you mix in one indicator two measures and it is not possible to separate effect of robustness and accuracy.
Tables 1 and 2. Why column tau does not contain tau? It is necessary to present tau. You can add additional column for used quality measures but tau must be presented. After long thinking I understand that tau are columns of table. It is necessary to reformat table and put tau in row before values of tau. Instead of tau you can use “quality measure” of “indicator”. You described that you measure quality by correlation between “between the estimated and true canonical variables”. Why do you include unrelated to this value \rho(a_1,b_1)? It must be removed because only contaminated table. You can form one more table to compare degree of correlation.
Line 312. I think reference should be number in [] but not [Frets].
Lines 311-323. Data description is omitted. From presented text I can guess that dataset contains length and breadth of head for the first (X) and the second (Y) son. And our goal to recognise sones of one family. Unfortunately, it is only my guess. Real life problem is not formulated. There is no interpretation of solutions from this problem point of view.
Line 317. Interpretation of points in figure 2 is absolutely unclear. What these figures mean?
Line 328. Hat is N_2 distance?
Figure 2. Why y_2 is located inside of right figures?
Line 337. I should stress that authors measure ability to find “pure solution” but not robustness. Even for used strange definition of robustness authors demonstrated small improvement of robustness with significant loss of accuracy.
I also should suggest to consider alternative measure of “nonlinear relationship”. You can compare two distributions of canonical variables directly through Kolmogorov-Smirnov test. I am not sure that this approach will be successful but it looks like reasonable generalisation. I do not insist in consideration of such approach and added it simple as suggestion for future generalisation.
The main drawback of paper – absence of description of problem under consideration.
Round 3
Reviewer 3 Report
I should repeat the same question – what is problem under consideration. Authors cannot formulate any reasonable problem. According to authors answer “In their context, the two groups of variables under consideration were hydrological variables and meteorological and/or graphical characteristics of the watersheds and their non-linear relationship depended essentially on the physiographic characteristics of the watersheds.” Now I can understand the nature of two sets of variables. Unfortunately, there is no question to answer. What I should to do? If I know both sets of random variables, then I really have one wider dataset. Why I should consider these two sets separately? I should say that authors should start from real life problem.
For example, we have “standard” image of many objects and we have “non-standard” images of the same objects. We separately converted images into some sets of features. Then we found canonical vectors, which reveal dependence between features of “standard” and “non-standard” images. On the set of canonical variables, we created classifier, which recognised images of one object as “the same” and other objects as “different”.
After formulation of this problem, we can play with different versions of CCA but all comparison MUST be presented for original problem: how successful was classifier for CCA, ICCA and for RPCCA. Otherwise, all comparison are simple play with small statistical toy.
I am not sure, that formulated above problem is reasonable for proposed technique. Authors should describe problems, for which proposed methodology is better than previously used. I do not think that it is reasonable to read paper further, because we do not have problem and, as a result, we do not have measure of success/fail.
Technical comments
I found strange statements such as “seasonal climate forecasting”. I almost sure that it means “seasonal weather forecasting” because climate is more general notion which does not depend on season.